# Mapping genomic and transcriptomic alterations spatially in epithelial cells adjacent to human breast carcinoma

Moustafa Abdalla[1,2,3,4], Danh Tran-Thanh[2], Juan Moreno[2], Vladimir Iakovlev[2], Ranju Nair[1], Nisha Kanwar[1,2], Mohamed Abdalla[2], Jennifer P.Y. Lee[2], Jennifer Yin Yee Kwan[2], Thomas R. Cawthorn[1,2], Keisha Warren[1], Nona Arneson[1], Dong-Yu Wang [1], Natalie S. Fox[5,6], Bruce J. Youngson[2,7], Naomi A. Miller[2,7], Alexandra M. Easson[8], David McCready[8], Wey L. Leong[8], Paul C. Boutros [5,6,9] & Susan J. Done [1,2,5,7]

Almost all genomic studies of breast cancer have focused on well-established tumours because it is technically challenging to study the earliest mutational events occurring in human breast epithelial cells. To address this we created a unique dataset of epithelial samples ductoscopically obtained from ducts leading to breast carcinomas and matched samples from ducts on the opposite side of the nipple. Here, we demonstrate that perturbations in mRNA abundance, with increasing proximity to tumour, cannot be explained by copy number aberrations. Rather, we find a possibility of field cancerization surrounding the primary tumour by constructing a classifier that evaluates where epithelial samples were obtained relative to a tumour (cross-validated micro-averaged AUC = 0.74). We implement a spectral co-clustering algorithm to define biclusters. Relating to over-represented bicluster pathways, we further validate two genes with tissue microarrays and in vitro experiments. We highlight evidence suggesting that bicluster perturbation occurs early in tumour development.

[1] Campbell Family Institute for Breast Cancer Research, Princess Margaret Cancer Centre, University Health Network, Toronto, ON, Canada M5G 2M9. [2] Department of Laboratory Medicine and Pathobiology, University of Toronto, Toronto, ON, Canada M5S 1A1. [3] Department of Biochemistry, University of Toronto, Toronto, ON, Canada M5S 1A8. [4] Department of Physiology, University of Toronto, Toronto, ON, Canada M5S 1A8. [5] Department of Medical Biophysics, University of Toronto, Toronto, ON, Canada M5S 1L7. [6] Informatics & Biocomputing Program, Ontario Institute for Cancer Research, Toronto, ON, Canada M5G 0A3. [7] Laboratory Medicine Program, University Health Network, Toronto, ON, Canada M5G 2C4. [8] Department of Surgical Oncology, Princess Margaret Cancer Center and University of Toronto, Toronto, ON, Canada M5G 2M9. [9] Department of Pharmacology & Toxicology, University of Toronto, Toronto, ON, Canada M5S 1A8. Correspondence and requests for materials should be addressed to S.J.D. (email: sdone@uhnres.utoronto.ca)

Numerous aberrations have been reported in histologically normal epithelium adjacent to breast tumours, including loss of heterozygosity (LOH), allelic imbalances[1–7] and transcriptomic alterations[8–13]. These likely represent some of the earliest genetic alterations in breast carcinogenesis as well as predisposing factors. However, alterations in normal epithelium have been elucidated relative to normal breast tissue from other

patients[8–10] or with bulk extracted tissue[11] making it impossible to confidently distinguish the earliest changes. Others have suggested that presence of contaminating tumour cells beyond the invasive tumour margin, rather than field cancerization (i.e. acquired molecular perturbations over a geographic region surrounding the tumour), may be the cause of local recurrence[14]. Here, we develop a map of alterations that occur in breast

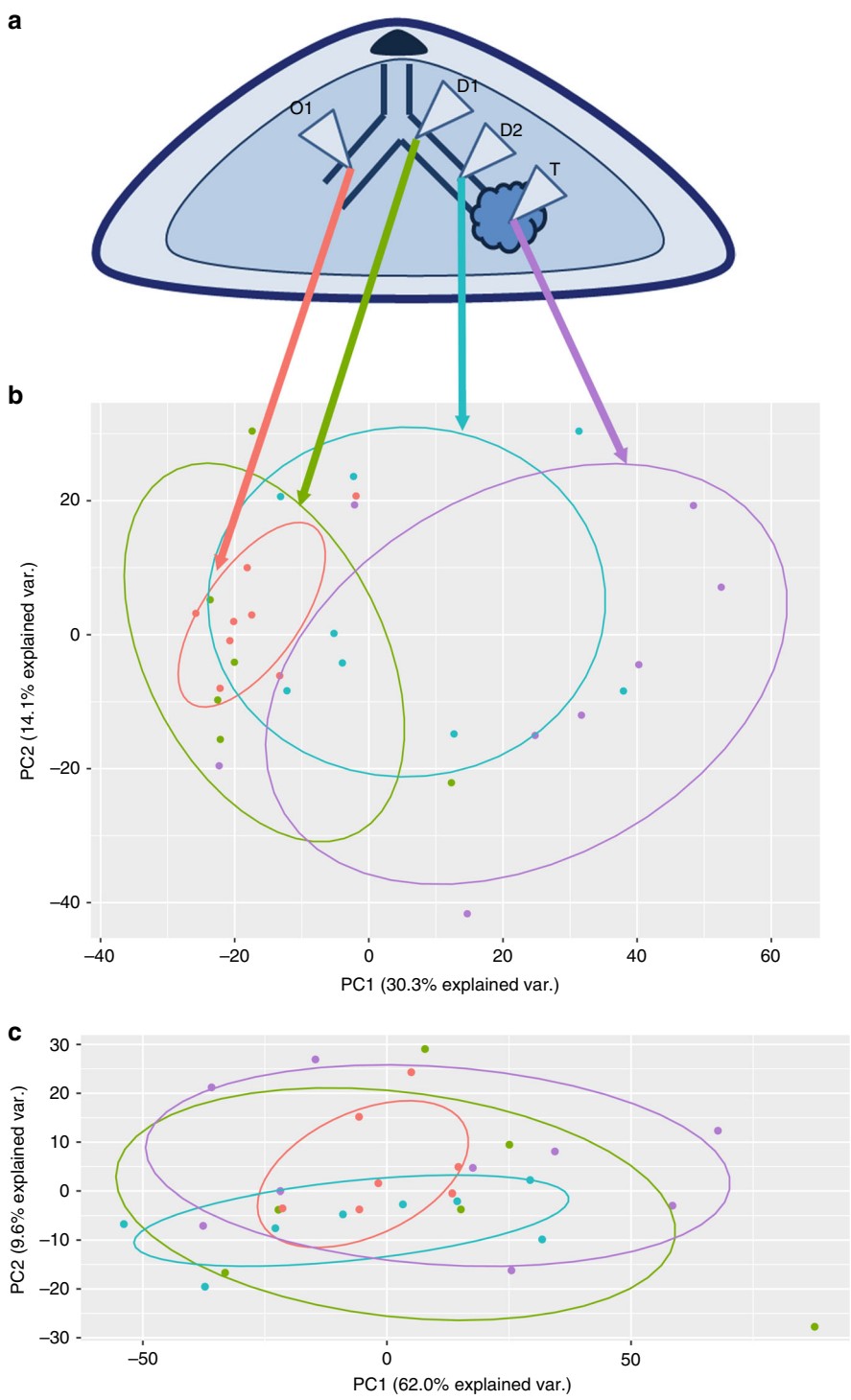

**Fig. 1** The first principal component of gene expression directly corresponds to proximity to tumour. **a** Schematic of the relative locations of all epithelial sample extractions, with the nipple and tumour highlighted. Samples D1 and D2 were obtained along the duct, approaching the tumour. Sample O1 is an epithelial sample from a breast duct on the other side of the nipple. **b** Biplot of the first two principal components of the expression matrix, limited to top 10% most varied genes. This proximity-based separation between epithelial samples is observed with the biplot including all genes, but is clearest here. **c** Biplot for the corresponding aCGH data; with no proximity-to-tumour separation in either of the first two principal components. PC, principal component

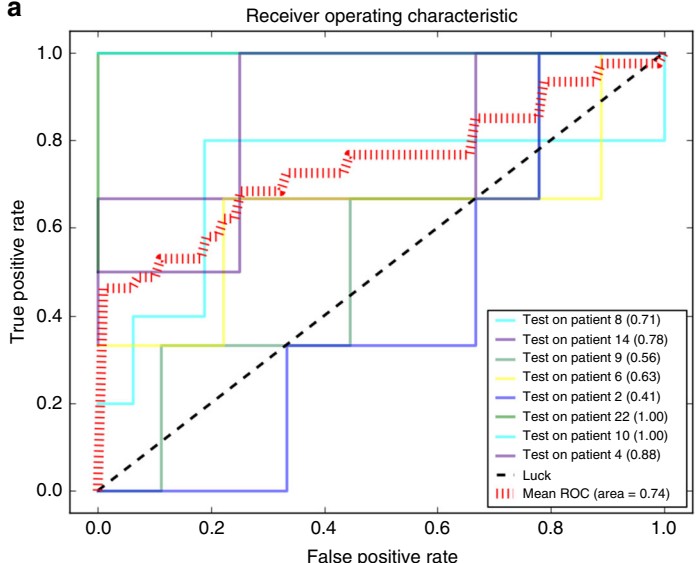

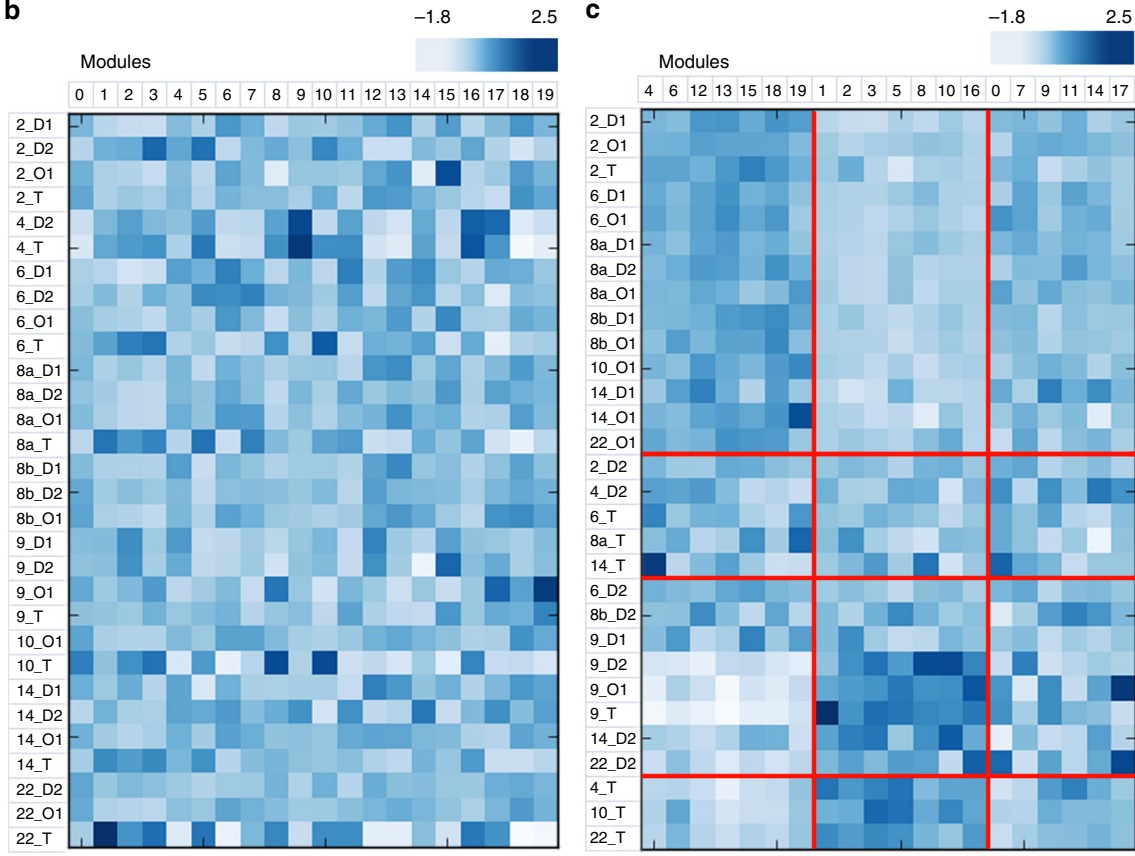

**Fig. 2** Expression-based classifier can identify where the epithelial samples were obtained from relative to the tumour. **a** Five-fold cross validation, depicting the receiver-operating curve (ROC) response of using each patient as a test; mean micro-average AUC = 0.74. **b** Original unclustered data set: columns depicted modules of genes identified through a hierarchical-clustering based gene agglomeration approach and rows corresponding to each epithelial sample. **c** Spectral co-clustering reveals biclusters of samples and modules that are both downregulated (modules 4, 6, 12, 13, 15, 18 and 19) and upregulated (modules 1, 2, 3, 5, 8, 10 and 16) in tumours and adjacent-to-tumour epithelium. All genes in these modules were selected a priori as univariately informative of the proximity-to-tumour label (top 30% of all genes; see Text). Clustering was done to further reveal structure within these univariately informative genes. Modules that are downregulated are negatively correlated with proximity to tumour (Table 1), and similarly, modules that are upregulated are positively correlated with proximity to tumour (Table 1). The sampling clustering has a Rand index of 0.21 with the distance-based grouping of the samples; the bottom three sample clusters are largely composed of tumour (T) and adjacent-to-tumour epithelial samples (D2). In contrast, the top cluster is composed largely of distant and contralateral duct epithelial samples (D1 and O1, respectively)

epithelium located at various distances from cancers in 8 patients, using copy number aberrations (CNA) and mRNA profiling, and microdissected under visual inspection. We evaluated how much of the variance in mRNA abundance, after adjusting for baseline levels, can be explained by copy number variation with increasing proximity to the tumour. We subsequently constructed a multi-class classifier that could identify from where the epithelial sample was obtained relative to the tumour; presenting evidence for field cancerization surrounding the primary tumour. Enrichment analysis of selected biclusters, defined using a spectral co-clustering algorithm, identifies pathways that may be increasingly dysregulated with proximity to tumour. Pathway analysis and enrichment of somatic mutations commonly present in breast cancer suggests these perturbations were not a consequence of paracrine influence from the primary tumour. Taken together, our analyses indicate that tumour initiation may not be driven by CNAs and that expression data points to an altered field surrounding the tumours. Identifying such transcriptomic alterations, preceding tumourigenesis, would allow us to better understand carcinoma development as well as develop new screening and treatment approaches.

## Results

**Generation of spatial-omic data.** To perform a spatial mapping of samples obtained at increasing distances from the tumour, we studied eight patients undergoing mastectomy for carcinoma. During surgery, prior to removal of the breast, a ductoscopy procedure was performed using a 0.7 mm mammary ductoscope. The duct leading to the tumour was identified by visual inspection. Methylene blue dye was injected to identify the involved duct. Immediately after the mastectomy, epithelial tissue sampling was performed. Two samples were taken along the duct between the tumour and the nipple (previously dye stained; D1 and D2) and one from a duct on the other side of the nipple within the same breast (as a normal control; O1). Six samples were defined as outliers using a sum of standard error boxplot (Methods section) and were excluded from subsequent analyses. The epithelial sample closest to tumour (D2) presented with mixed histology: with some normal ($n = 5$) and some atypical ductal hyperplasia (ADH)/ductal carcinoma in situ (DCIS; $n = 3$). D1 and O1 were all histologically normal. A sample from the tumour was also obtained (T; schematic in Fig. 1a). The clinico-pathological features for this subset are summarised in Supplementary Data 1. To determine the degree and extent of genetic and molecular alterations along the mammary ducts leading to the breast cancer, the samples obtained along the duct up to and including the tumour were then microdissected to ensure >90% epithelial cells, absence of morphologically malignant cells and subject to copy number and mRNA abundance profiling.

**CNAs explain little spatial heterogeneity in mRNA abundance.** We identified increasing CNAs with proximity to tumour (Supplementary Data 2). To assess whether these genomic changes had concomitant transcriptomic perturbations, for each sample from the duct leading to the tumour (D1, D2) and tumour (T), we performed a regression analysis using the epithelium sample from the opposite duct (O1) as the predictor of mRNA abundance. The residuals from this analysis represented a transformed variable that was used for subsequent modelling; residualising gene expression on the background sample leaves the part of gene expression that is unrelated to natural variation and only affected by the proximity to tumour. A simple regression model demonstrates that CNAs explain almost none of the variance (equivalently, perturbations) observed in the expression data along the tumour duct (Supplementary Table 1). Thus, CNAs are not

informative regarding epithelial proximity to tumour. Notably, however, in some patients, CNAs can explain a small part of the expression perturbations observed in the tumour itself (T). In select patients, we detected aberrations across all biopsies suggesting that the CNAs arose during mammary gland development (similar genes are perturbed across all samples as tabulated in Supplementary Data 2). However, as evidenced from the models, CNAs cannot explain the gradual perturbation in mRNA abundance observed with decreasing distance to tumour, suggesting that their effects on the expression profile may be limited. Notably, however, we found that the first principal component of the gene expression matrix separated epithelial samples by their proximity to the tumour (Fig. 1b).

**Expression classifier can identify epithelial sample locations.** Having observed that the first principal component of the abundance matrix corresponds to distance from tumour, we were interested in whether mRNA abundance data was sufficiently informative about tumour proximity. We constructed a multi-class one versus rest classifier, extending a C-support vector classification with a linear kernel. To quantify and study the output of the classifier, we extended ROC curves by considering each element of the label indicator matrix as a binary predication (i.e. microaveraging). With this simple classifier, we were able to obtain an 8-fold cross-validated AUC of 0.74 (each fold was tested on one patient (i.e. a set of O1, D1, D2 and T from one individual) and trained on all remaining patients) (Fig. 2a). Classifier performance was similar between patients with histologically normal D2 (adjacent-to-tumour epithelial samples) and patients with ADH/DCIS in D2 samples (patients 4, 9 and 14). Thus, mRNA abundance is sufficiently informative about proximity to tumour. More generally, this is evidence of field cancerization—there exists transcriptomic perturbations surrounding the primary tumour that can be detected within expression data using this classifier. The discriminative ability of the classifier further suggests that perturbations observed in adjacent-to-tumour epithelium are likely shared within the cohort (i.e. the classifier model is generalisable and is not overfitting to single patient-specific patterns). In other words, this is evidence that perturbations in epithelial samples that are proximal to the tumour (i.e. adjacent-to or within the same duct) are shared across patients. This, alongside the mutation analysis presented below, suggests that the cancerization is independent of tumour paracrine influences.

**Co-clustering identifies genes that correlate with distance.** Having demonstrated that mRNA abundance is informative with respect to epithelial distance from tumour, we were interested in correlating this information with biological and pathway perturbation. More concretely, we sought to identify biclusters with expression values higher (or lower) with samples closer to the tumour than those further away, using spectral co-clustering. These biclusters will allow us to delineate which features (genes) of the transcriptome are informative of epithelial distance to tumour. For computational feasibility, we limited the abundance matrix to the top 30% of most informative genes using the uni-variate non-parametric mutual information estimate implemented for the multi-label classifier. We agglomerated genes into 20 clusters representing modules of genes with the minimum variance (Ward linkage criterion and Euclidean affinity); 20 clusters maximised the match between the sample clustering and the distance annotation label using the Adjusted Rand Index (ARI). ARI is a chance-adjusted similarity measure between two groupings[15]. More importantly, no assumption is made on the underlying structure and it is easy to interpret; an ARI score of 0

**Table 1 Tabulated summary of Pearson correlation coefficients between the modules identified with spectral co-clustering and proximity to tumour, to three significant digits**

| Modules | Pearson's *r* | *p*-value | Bonferroni adjusted | Significant |
|---|---|---|---|---|
| 0 | −0.355 | 0.054 | 1.075 | |
| 1 | 0.597 | 0.000 | 0.009 | ** |
| 2 | 0.612 | 0.000 | 0.006 | ** |
| 3 | 0.672 | 0.000 | 0.001 | ** |
| 4 | −0.412 | 0.023 | 0.460 | |
| 5 | 0.542 | 0.002 | 0.037 | ** |
| 6 | −0.654 | 0.000 | 0.002 | ** |
| 7 | −0.228 | 0.226 | 4.519 | |
| 8 | 0.460 | 0.010 | 0.204 | |
| 9 | 0.157 | 0.406 | 8.118 | |
| 10 | 0.556 | 0.001 | 0.027 | ** |
| 11 | 0.101 | 0.595 | 11.908 | |
| 12 | −0.478 | 0.007 | 0.145 | |
| 13 | −0.608 | 0.000 | 0.007 | ** |
| 14 | 0.173 | 0.359 | 7.181 | |
| 15 | −0.383 | 0.036 | 0.723 | |
| 16 | 0.632 | 0.000 | 0.003 | ** |
| 17 | −0.176 | 0.351 | 7.016 | |
| 18 | −0.672 | 0.000 | 0.001 | ** |
| 19 | −0.511 | 0.004 | 0.074 | |

Significant modules are denoted with **. Full list of genes and modules is available in Supplementary Data 3. All genes in these modules were univariately informative of the proximity-to-tumour label

indicating random assignment and a maximum score of 1 indicating perfect overlap. The 20 clusters maximised overlap with the true proximity-to-tumour annotations (ARI = 0.21; Fig. 2b). As spectral co-clustering treats the expression matrix as a bipartite graph (samples and modules representing two sets of vertices), each sample and each module belong to one bicluster with higher expression than for any other sample or module (Fig. 2b). In other words, biclusters represent sets of genes that are upregulated in a subset of samples; rearrangement of the rows and columns to make bicluster partitions contiguous reveals high expression values along the diagonal (Fig. 2b). It is important to note that all genes in these modules were a priori selected as informative of proximity-to-tumour label (using the univariate non-parametric mutual information estimate). Clustering further reveals structure within these univariately informative genes. To further quantify the correlation of these modules with proximity to tumour, we calculated the Pearson correlation between the module and proximity to tumour (Table 1) and identified nine significantly correlated modules (after conservative Bonferroni adjustment). Three modules (6, 13 and 18) negatively correlated with proximity to tumour and six modules (1, 2, 3, 5, 10 and 16) correlated positively with proximity to tumour (Table 1).

**Transcriptomic alterations are a consequence of cancerization**. Having noted increasing transcriptomic dysregulation with increasing proximity to tumour, we were interested in establishing whether this mRNA abundance perturbation is a consequence of contaminating tumour cells, paracrine (secretory) influences from the tumour or is evidence of field cancerization. As all specimens were manually microdissected by a pathologist and obtained under direct vision, the likelihood of tumour cell contamination was low. To distinguish between paracrine and field cancerization effects, we noted that if modules that correlate with proximity to tumour are drivers of tumourigenesis, we would expect a larger number of mutations (such as missense or splice alterations) in module genes from breast cancer patients compared to background genes that do not belong to these significantly correlated modules. If paracrine effects predispose cells to increased mutations, all genes should be affected equally (i.e. no enrichment). To test this hypothesis, we collected mutation data from 8 different breast cancer cohorts, including The Cancer Genome Atlas[16], [17], METABRIC[18], and data sets from the Broad[19], Sanger[20], and British Columbia Cancer Research Centre[21] (curated by the cBIO Cancer Genomics Portal[22]). We limited our analyses to genes that were both present on the microarray platform and in at least one cohort. We noted that genes in modules that correlate (positively and) significantly with proximity to tumour were more likely to be mutated—compared to all other bicluster genes—in breast cancer patients across all eight data sets ($\chi^2$ test; Yates $p < 0.05$; odds ratio = 1.33; 95% confidence interval = (1.18, 1.51); Supplementary Table 2). A similar trend was noted for modules that negatively and significantly correlate with proximity to tumour ($\chi^2$ test; Yates $p < 0.05$; odds ratio = 1.18; 95% confidence interval = (1.01, 1.38); Supplementary Table 3). This suggests that the observed transcriptomic dysregulation is more likely a consequence of field cancerization rather than paracrine influence from the primary tumour.

With this genetic evidence in support of 'field' transcriptomic perturbation, we were subsequently interested in discovering the functional nature and clinical relevance of the modules significantly correlated with distance to tumour. No module stood out with respect to absolute magnitude of the strength of correlation with proximity to tumour. Thus, we elected to arbitrarily select two module containing genes for which we had tissue microarray (TMA) data: module 16 (containing *MSI2*; Pearson's $r = 0.63$; Bonferroni adjusted $p < 0.05$) and module 2 (containing *SPAG5*; Pearson's $r = 0.61$; Bonferroni adjusted $p < 0.05$). Full list of genes and modules is available in Supplementary Data 3.

**MSI2 is a novel member of the Wnt pathway**. Since module 16 is positively correlated with proximity to tumour ($r = 0.63$), we sought to determine the biological relevance of this mRNA module and assess whether there was evidence of changes in pathway activation relative to tumour proximity. The most statistically significantly over-represented pathway in this module relative to all others is formation of the "beta-catenin:TCF-transactivating complex" ($q = 5.67 \times 10^{-20}$), with 25/79 genes in this pathway that were abundance-profiled as being present in the 289-gene module (expectation is ~1 gene in this pathway). Complete lists of genes and enriched pathways are available in Supplementary Data 4 and 5, respectively. Thus, we hypothesised that the Wnt pathway may be a signalling driver of this module. A corollary of this is that other genes of this module may be involved with Wnt signalling through guilt by (coexpression) association[23], i.e. are novel components of the pathway. To partially validate that module is a Wnt signalling module, we selected a gene not previously associated with Wnt and studied its role in the pathway: *MSI2*. MSI2 has an established functional role in neural stem cell maintenance[24] and in the regulation of hematopoietic stem cell self-renewal[25]; *MSI2* has also been associated with other signalling pathways including Hedgehog and Notch[26].

Initially, we used activation of a (transcription factor)/LEF-1-dependent luciferase reporter construct (TOPFLASH) in MCF7 (a luminal breast cancer cell line) to assess the effect of MSI2 overexpression on Wnt signalling. Notably, we observed a significant 1.5-fold increase in luciferase activity (compared to control Flag transfections; paired Student's *t* test; $p < 0.05$; Fig. 3a). Subsequently, to elucidate the mechanism of action of

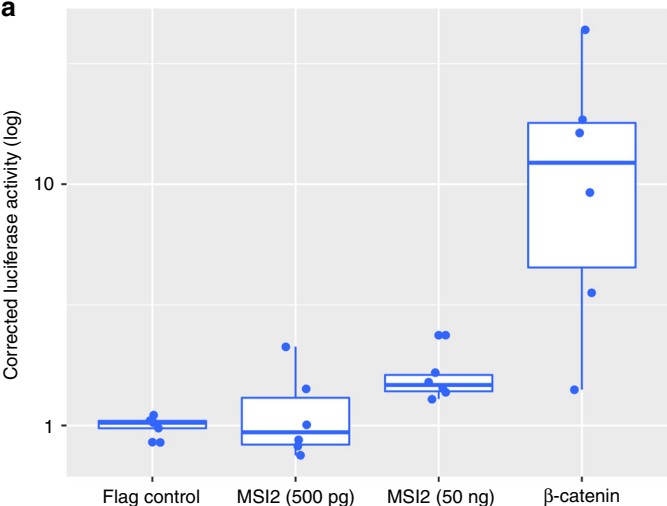

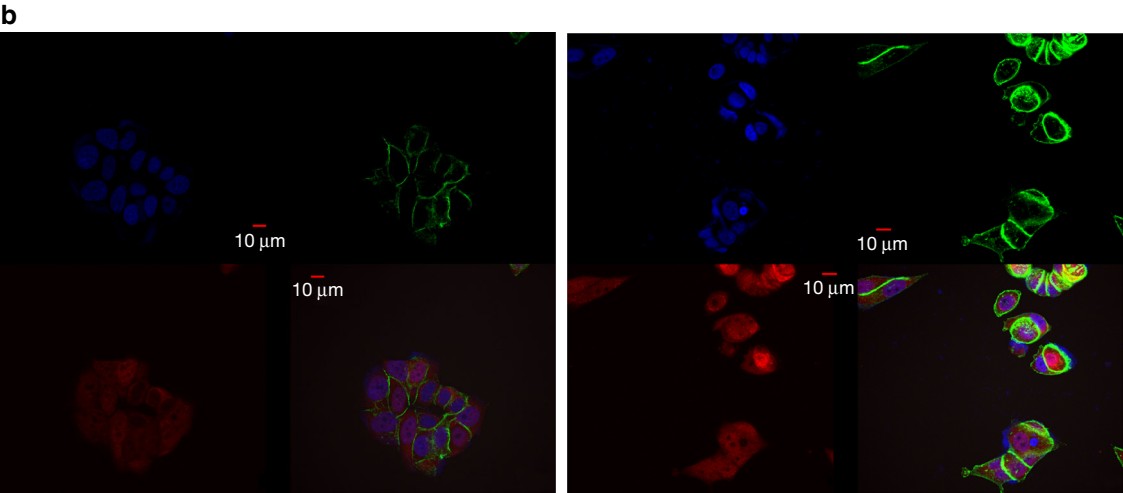

**Fig. 3** *MSI2* overexpression activates the Wnt cascade. **a** MSI2 overexpression results in a significant 1.5-fold linear increase in TOPFLASH Luciferase activity, relative to Flag control (paired Student's *t* test; $p < 0.05$; please note the log scale). **b** MSI2 overexpression mediates β-catenin localisation to the nucleus, with no change in the total intracellular β-catenin levels. Left panel: representative immunofluorescence image of control MCF7 (nucleus is highlighted with blue DAPI staining; β-catenin is stained green; and MSI2 is red). β-catenin is largely localised to the membrane and the cytosol. Right panel: representative immunofluorescence image of MCF7 MSI2 overexpression clones, with β-catenin localisation to the nucleus

MSI2, its effect on β-catenin was determined. Given the central role of β-catenin activity for changes in canonical Wnt signalling, we investigated whether MSI2 altered β-catenin translocation using immunofluorescence (IF). Increased MSI2 overexpression caused nuclear localisation of β-catenin in MCF7 (Fig. 3b), further corroborating MSI2—and more generally, the module's role—in the Wnt signalling cascade. Furthermore, the increasing expression of *MSI2* with proximity to tumour, alongside its effects on β-catenin translocation, is consistent with the module-specific enrichment of components involved in formation of the β-catenin:TCF-transactivating complex.

**MSI2 abundance is associated with tumour grade**. Having implicated MSI2 in the Wnt pathway and partially validating that the module is a candidate Wnt-driven module, we were interested in exploring whether MSI2 protein abundance correlated with other clinical traits. A consecutive series of 232 invasive breast cancers was evaluated using in-house TMAs (Supplementary Fig. 1) to assess the status of MSI2 protein abundance in a larger cohort of breast cancer patients. MSI2 protein expression was seen in both the cytoplasm and the nucleus. MSI2 protein

abundance was significantly negatively associated with ER and PgR status ($p < 0.05$; Methods section; Supplementary Fig. 1). Representative staining of the TMAs is shown in Supplementary Fig. 2.

To characterise the functional consequences of increased *MSI2* expression in vivo we used a transwell assay to evaluate migration. MSI2-GFP-expressing MCF7 cells exhibited a significant 104% increase in migration relative to control cells expressing GFP alone (95% confidence interval = 11.6% to 198.4% increase; Welch's *t* test; $p < 0.05$; Fig. 4a). The number of invasive cells also increased by 77% compared to the GFP controls in MCF7 (95% confidence interval = 23.6% to 130.8% increase; Welch's *t* test; $p < 0.05$; Fig. 4b). Similar results were observed in the MDA-MB-231 cell line over-expressing a MSI2-GFP fusion (Welch's *t* test; $p < 0.05$ for both; 193 and 291% increase for migration and invasion, respectively; Fig. 4c, d). The MSI2-knockdown MCF7 clones exhibited the reverse effect in the migration and invasion assays (Fig. 4e, f). The knockdown cell line was further assessed for proliferative capacity by the cell counts at different days. By day 5, MSI2-knockdown cells exhibited significantly decreased proliferative capacity, in contrast to the controls (Welch's *t* test; $p < 0.05$; Supplementary Fig. 3).

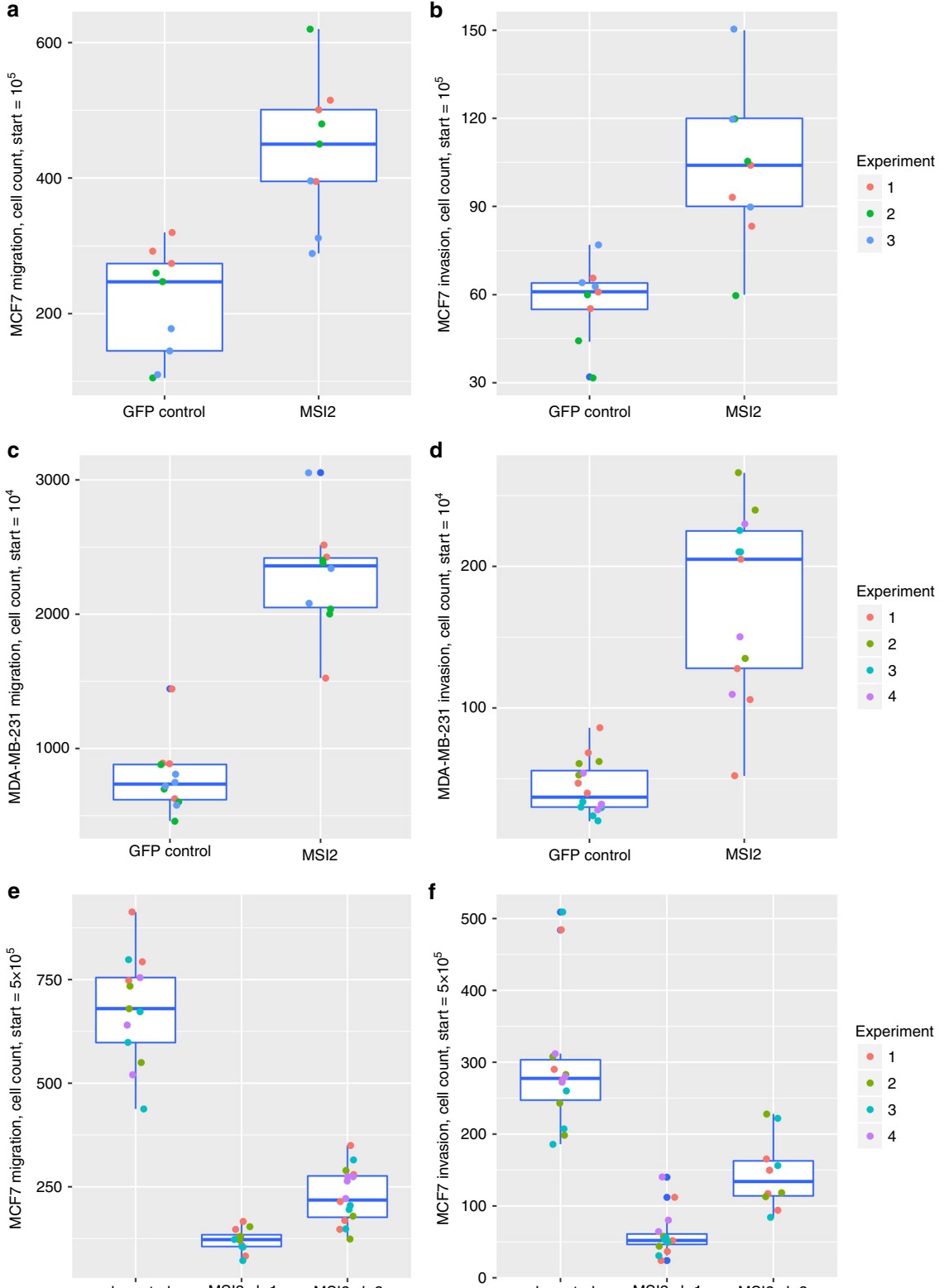

**Fig. 4** *MSI* overexpression increases migration and invasion of MCF7 and MDA-MB-231 MCF7-expressing GFP and MSI2-GFP were counted by Vi-cell-XR, and equally plated on transwells with and without Matrigel. Migration **a** and invasion **b** qualities were assessed by counting cells 48 h after initial plating. MSI2 overexpression causes an increase in migration and invasion (Welch's *t* test; *p* < 0.05). Box plots of all replicates within each experiment (*n* = 3) are depicted. MDA-MB-231-expressing GFP and MSI2-GFP were plated on transwells with and without Matrigel. 48 h after initial plating, the transwells were counted for migration **c** and invasion **d**. MSI2 increased both the migrative capabilities and invasive tendencies of MDA-MB-231 (Welch's *t* test; *p* < 0.05). As with the MCF7, box plots of all replicates within each experiment (*n* = 4) are depicted. Migration **e** and invasion **f** were also assayed in MCF7 shRNA control and shRNA MSI2, respectively. The knockdown clones presented the opposite effect, with a significant decrease in the migration and invasion ability of MCF7 (Welch's *t* test; *p* < 0.05). Proliferation assay results are summarised in Supplementary Fig. 3

**SPAG5 abundance is associated with tumour grade**. Analogous to our previous analysis, we were interested in whether any signal transduction pathways were over-represented in Module 2, a 371-gene module (all genes are listed in Supplementary Data 6). None were discovered; full over-representation analysis is available in Supplementary Data 7. Moreover, the module appears to be enriched in gene sets relating to oxidative phosphorylation, the electron transport chain, the proteasome and metabolism, more generally. This suggests that this module is enriched for genes necessary for cell survival and possibly, proliferation. Subsequently, we elected to study one gene from this module with increasing expression with proximity to tumour. In particular, we decided to further explore the role of SPAG5 protein in breast tumourigenesis (for which we had TMA data), as it has been previously implicated in the maintenance of spindle-pole integrity, efficient chromosomal alignment and cell proliferation[27]. SPAG5 has been shown to interact with the mitotic spindle apparatus and is expressed in most human cell lines and tissues[28, 29]. The protein produced by the gene is a homodimer, with a globular head domain that interacts to form aster-like structures, which attach to microtubules in the mitotic spindle. To further explore the role of this gene in breast cancer, FFPE TMA blocks consisting of invasive breast cancers from 234 patients, were analysed (Supplementary Fig. 1). Immunohistochemical staining with an anti-SPAG5 antibody revealed that tumour grade was significantly and positively ($p < 0.05$; Methods section) associated with SPAG5 protein levels.

## Discussion

Identifying genomic alterations in mammary epithelial cells that precede breast carcinoma will allow us to better understand carcinoma development and heterogeneity. By obtaining and profiling morphologically normal epithelial samples at various distances from the tumour (and a sample from the tumour itself), we were able to draft a spatial map of the genomic events and transcriptomic alterations that occur along the mammary duct (leading up to and including the tumour).

In particular, using aCGH, we confirmed that the region surrounding the tumour is remarkably unstable with heterogeneous and varied genetic alterations; consistent with previous reports of LOH events and allele imbalances in adjacent-to-tumour epithelium (compared to normal breast tissue from the same patient)[1, 2]. Furthermore, we have shown that genomic alterations (in the form of copy number alterations) do not explain much of the expression perturbation that is observed with proximity to tumour. Through principal component analysis of the gene expression matrix, we observed that the first transcriptomic principal component corresponds directly to proximity to tumour. As we discuss below, we showed that transcriptomic alterations in spatially informative coexpression modules are likely more important than the paracrine influence of the tumour and this perturbation may precede tumourigenesis. Furthermore, the enrichment of somatic mutations in genes that correlate with proximity to tumour suggests a non-paracrine mechanism underlying the dysregulation. If paracrine effects predispose cells to increased mutations, all genes should be affected equally (i.e. no enrichment). Furthermore, if a paracrine effect is at play, it is also unlikely that it will extend so far beyond the tumour as our samples were taken and that it would produce a similar effect regardless of the adjacent tissue type (fibrous, adipose, epithelial, etc).

To the best of our knowledge, this represents the first direct evidence of field cancerization surrounding the primary tumour. All current evidence of global gene expression dysregulation in adjacent-to-tumour (but not necessarily in the same duct system) epithelium depends on comparisons to external patient data sets (such as breast tissue from patients with mammary hypertrophy undergoing reduction mammoplasty, from cohorts with atypical hyperplastic proliferative lesions, or simply just healthy individuals without cancer unmatched for clinical traits)[9, 10]. Other studies, that demonstrate widespread differential gene expression between patient-matched breast normal tissue, have confounding variables: such as analysis of bulk tissue of various types instead of more pure microdissected epithelial samples[11]. Unlike previous studies, we have further demonstrated that gene expression can identify from where the epithelial sample was obtained relative to the tumour (with a cross-validated average AUC = 0.74). This is the strongest evidence that epithelial samples surrounding the tumour differ from epithelial samples from the contralateral duct within an individual. The discriminative ability of the classifier suggests that these differences (or perhaps, perturbations) are shared across the cohort. Combined with the inability of copy number alterations to explain these transcriptomic patterns, this suggests that other forms of dysregulation (such as single-nucleotide variation, miRNA or epigenetic modifications) may be the primary drivers of initiation and progression in early tumour development.

Subsequently, using spectral co-clustering, we identified biclusters (i.e. coexpressors) with perturbations that may be among the first molecular alterations shared by breast cancer patients, preceding common aberrations detected by aCGH. Genes in these biclusters were a priori selected as informative for proximity-to-tumour label (prior to the co-clustering). We demonstrate that the genes in modules with significant correlation to proximity to tumour were more likely to be mutated in breast cancer patients (compared to background genes in non-significant modules), suggesting that this perturbation is a consequence of field cancerization (rather than paracrine influence from the primary tumour). As these genes are enriched for somatic mutations in tumours (i.e. they are more likely to be mutated), this further suggests a functional role for many of these genes in tumour development and progression. Furthermore, in an analysis by Rheinbay et al.[30], deep sequencing of 360 primary breast cancers revealed six promoters (TBC1D12, LEPROTL1, ZNF143, RMRP, ALDOA and FOXA1) that contained single-site mutational hotspots (three or more mutations at a single site). Three genes (*ZNF143, ALDOA* and *LEPROTL1*) were among the top 30% of genes that are univariately informative of the proximity-to-tumour label. Two genes (*TBC1D12* and *RMRP*) were not on the expression platform; *FOXA1* was not informative of the spatial label for the samples and was the only gene (among the 4 present on the platform) with the mutational hotspot not located in or adjacent to the 5′ untranslated region. Two genes (*ALDOA* in module 5 and *LEPROTL1* in module 2) also belonged to modules that significantly and positively correlated with proximity to tumour. Thus, non-coding single-nucleotide mutations may be responsible, in part, for the observed field cancerization.

Moreover, other genes in high-correlation modules have been previously implicated in early carcinogenesis. For instance, *MAPK14* (also known as p38) is present in module 2 and Gauthier et al.[12] have noted increased activated phospho-p38 staining intensity in the adjacent normal epithelium to DCIS compared with normal epithelium from reduction mammoplasty specimens. This is consistent with the positive correlation of module 2 with proximity to tumour. MAPK14 acts to stabilise the immediate-early gene *COX2*, also implicated in tumourigenesis[12]. We found associations for two of these coexpression modules with signalling cascades, and correlated two genes with clinical covariates. In particular, we noted the presence of *MSI2* in module 16, which was positively correlated with proximity to

tumour and enriched for genes relating to the formation of the beta-catenin:TCF-transactivating complex (of the Wnt signalling cascade). Notably, *MSI2* is located at 17q23.2-23.3 (55.1–55.9 Mb) [24]; part of an amplicon (17q22-24.2) whose gain is a predictor of invasion in DCIS and of nodal metastasis in invasive duct carcinoma (IDC) [31]. We were able to directly implicate *MSI2* in Wnt activation, with overexpression leading to β-catenin localisation to the nucleus and increased luciferase activity (i.e. pathway activity) using a TCF/LEF-1-dependent luciferase reporter construct. This suggests that Wnt signalling may be one of the earliest changes in breast cancer, as well as tumour development and progression. Studies in neuroblastoma and ovarian cancer also revealed amplification of the MSI2 genomic region, suggesting a role for both gene and genomic interval in oncogenesis and tumour progression [32–34]. More generally, this hints at the possibility that Wnt signalling may be implicated in early changes in other tumour types as well. Looking at another module that is both significantly and positively correlated with proximity to tumour (module 2), we note the presence of SPAG5, which positively correlates with tumour grade. Module 2 includes genes whose abnormal function has been directly associated with breast cancer, such as MAPK14, BRCA1, CDH1, HIF1A, CDKN2A and other members/regulators of the CDK family. It is important to note that membership to a module that positively correlates with proximity to tumour may be indicative of either a function in early tumourigenesis (e.g. MAPK14) or a protective role in response to the increased stress and dysregulation of the cellular environment (as genes in these modules are enriched for somatic mutations in tumours with both gain-of-function and loss-of-function consequences).

Here, in summary, we presented the first spatially comprehensive CNA and mRNA characterisation of morphologically normal epithelium from primary breast cancer patients of multiple molecular sub-types. Our results emphasise previous findings that Wnt signalling may be one of the earliest changes in breast cancer and we identify a new gene involved in the Wnt pathway. Our discovery of SPAG5 as an early gene suggests the presence of increasing chromosomal alterations with decreasing distance to tumour; consistent with our CNA observations. This study, however, did not address whether common epigenetic field effects, single-nucleotide non-coding variation, miRNA regulation or shared telomere dysfunction may influence the observed transcriptomic perturbation with increased proximity to tumour. However, there is evidence in the expression data indicating field alterations and that CNAs do not readily explain this field cancerization. Our results have important implications for cancer screening and prevention and could lead the way to new therapeutic approaches and targets.

## Methods

**aCGH and gene expression dataset preparation.** Fifteen patients undergoing mastectomy for biopsy-proven carcinoma were recruited at the Princess Margaret Cancer Center, University Health Network (Toronto, ON, Canada) from 2005 to 2008. Informed consent was obtained after the nature and possible consequences of the study were explained. Inclusion criteria were defined as tumours at least 2 cm in diameter as measured by imaging, and located at least 3 cm from the nipple. Institutional REB approval was obtained prior to patient recruitment. Clinico-pathological data was obtained for all patients for whom all eight samples (four expression and four aCGH) passed quality control (QC) at the microarray centre ($n = 8$). During surgery, prior to removal of the breast, a ductoscopy procedure was performed using a 0.7 mm mammary ductoscope (MF2-707, MD Fibertech Co, Japan). Details of tissue sampling are described in the main text. All tissues obtained were bisected, with one half snap frozen and the other half formalin-fixed and paraffin-embedded (FFPE). Normal lymph node tissue was also obtained for control DNA. Snap frozen tissues were sectioned at 8 μm thickness using a microtome and lightly stained with hematoxylin. Histological identification of tumour and normal ducts was confirmed by a pathologist. Needle-microdissection was performed by a pathologist with a stereoscopic dissecting microscope for both tumour and ducts to ensure minimal stromal contamination of epithelial cells.

DNA and RNA were extracted from microdissected tissue using the Qiagen All-prep RNA/DNA Micro Kit (Qiagen, Mississauga, ON, Canada).

**Array comparative genomic hybridisation and analysis.** Whole genome amplification was performed using the single-cell comparative genomic hybridisation protocol as described previously [35]. Amplified DNA was labelled using the Agilent Genomic DNA ULS Labeling Kit and hybridised using the aCGH Hybridization Kit on Agilent Human 244k arrays (Agilent Technologies, Mississauga, ON, Canada) as per manufacturer's standard protocol. Pooled lymph node tissue from patients was used as controls and processed in a similar fashion. Raw microarray image files were extracted using Agilent Feature Extraction Software (v9.5). Data were imported into Agilent Genomic Workbench v7.0.4.0 (Agilent Technologies, Santa Clara, CA, USA) for QC analysis. QC metrics were generated for all samples; none were excluded from subsequent analyses. aCGH data were subsequently analysed and visualised using rCGH v1.2.2 (bioconductor v3.3); package code and resources were modified to work with hg17 (genome build of data).

**Expression microarray protocol and analysis.** Global cDNA Whole-Human Genome 4 × 44 K Microarray (Agilent) gene expression profiling was performed on the same tissue specimens previously mentioned, as described elsewhere [36]. Briefly, 100 ng of mRNA were reverse transcribed using Superscript II reverse transcriptase (Invitrogen Canada, Burlington, ON) while incorporating Cy3-dCTP or Cy5-dCTP (NEN, Boston, MA, USA). Raw microarray image files were extracted using Agilent Feature Extraction Software (v9.5). Microarray data were pre-processed and normalised using agilp v3.4.0 (bioconductor v3.3); only probes that matched to a gene symbol were included. In case of multiple ID mappings, the average was calculated between the matching probes, with no filtering on variance or expression threshold. Six samples (4_D1, 4_O1, 8b_T, 10_D1, 10_D2 and 22_D1) were excluded, after QC, and were defined as outliers using a sum of standard error boxplot.

**Bioinformatic analyses.** To quantify the variance explained by CNAs, linear regression was performed with R v3.2.2. After filtering based on intersection of genes present in both the processed mRNA abundance and aCGH data, residuals were obtained for each sample along duct leading to tumour (from the expression—contralateral sample equation) and a linear regression model was built in the form of residuals—aCGH data. Both residuals and aCGH came from the same epithelial sample (i.e. at the same distance from the tumour). Principal component analysis of the expression matrix was completed using base functions of R v3.2.2 and visualised using ggbiplot (freely available at, https://github.com/vqv/ggbiplot/). We implemented the multi-label classifier using scikit-learn 0.18 using OneVs-RestClassifier with a C-Support Vector Classification linear-kernel estimator. ROC curves were drawn by considering each element of the label indicator matrix as a binary prediction (microaveraging). Spectral co-clustering was also performed using scikit-learn 0.18, with feature agglomeration, incorporating a ward linkage criterion and a Euclidean affinity. The number of clusters was optimised to increase overlap of sample biclusters with known distance annotations, as quantified using the adjusted Rand score.

**MSI2 overexpression.** A total of 5 μg MSI2-GFP and GFP vectors (pCMV6-MSI2-AC-GFP and pCMV6-AC-GFP from Origene, Rockville, MD, USA) were transfected into MCF7 and MDA-MB-231 cells (source: ATCC) using endofectin transfection reagent (Genocopoeia, Rockville, MD, USA) according to the manufacturer's instructions. Expression was determined by fluorescence microscopy, western immunoblotting and qRT-PCR. Cell lines were authenticated and quality controlled at ATCC.

**Migration and invasion assays.** For the migration and invasion assays, stable (overexpression) clones were selected with G418 (2.3 mg/ml) for 5–10 passages (for both MCF7 and MDA-MB-231). Stable MSI2-knockdown clones were established using shRNA (Origene) using a lentiviral transfection in 293 T cells, followed by transduction and selection with puromycin (0.6 μg/ml). Knockdown of MSI2 expression in individual clones was confirmed by RT-PCR. For invasion and migration assays, 24-Transwell plates (8-μm pore size; BD Biosciences, Franklin Lakes, NJ, USA) were coated with Matrigel (BD Biosciences) or without a coating, respectively. $10^5$ cells were plated on the upper side of the inserts, and allowed to migrate or invade to the underside of the insert over 48 h. Inserts were fixed and stained. The total number of cells on the underside of the insert was counted at the microscope. Each experiment was performed in triplicate. Welch's *t* test was used to assess significance.

**Immunofluorescence staining.** MCF7 cells were grown on 22 mm round collagen type I coated coverslips (BD Biosciences) until 70% confluent. The cells were then fixed with 4% paraformaldehyde solution for 15 min at room temperature. Fixed cells were permeabilised by washing twice in PBS with 0.01% Tween20 for 5 min. Blocking was carried out with 1% BSA in the above PBS/Tween solution for 30 min at room temperature. Cells were then incubated in a mixture of the two primary antibodies (rabbit against MSI2 and mouse against β-catenin from BD transduction

lab #610153; 1:100 dilution in 1% BSA/PBS/Tween overnight at 4 °C). Following three 3-min washes in ice cold PBS, fluorescently labelled secondary antibodies were applied for one hour. Using FV10-ASW version 2.1c imaging software, each image was captured sequentially with four channels (Cy3, FITC, DAPI and DIC) to eliminate crosstalk between channels. All images were captured at ×60 magnification.

**Luciferase assay**. MCF7 cells were transfected with 400 ng of the TOP Luciferase reporter transgene, 40 ng of a constitutive renilla luciferase transgene and 500 pg or 50 ng of flag-MSI2 using Lipofectamine LTX with Plus Reagent (Life Technologies) according to the manufacturer's instructions. Cell lysates taken after 48 h and luciferase activities were determined using the Dual-Luciferase Reporter Assay System (Promega) according to manufacturer's instructions. Three successful assays ($n = 3$ replicates each) were completed demonstrating that an increase in Wnt signalling corresponded with MSI2 overexpression. However, only two were plotted and analysed (one way ANOVA; paired Student's $t$ test) as the FOP negative control was unsuccessful for 1 of the 3 trials.

**MSI2 TMA**. Immunohistochemical analysis of MSI2 expression in breast cancer cells was performed. Briefly, slides sectioned from an in-house (UHN) TMA were dewaxed in xylene and rehydrated through alcohol to water. The TMAs consisted of a consecutive series of FFPE invasive breast carcinomas ($n = 234$ but two cases were technically unsatisfactory) collected from a single institution (UHN) in 2006. Endogenous peroxidase was blocked with 3% hydrogen peroxide for 10 min. Microwave antigen retrieval was carried out in 10 mM citrate buffer pH6. Endogenous biotin was blocked with Vector's biotin blocking kit and slides were then incubated with MSI2 (Millipore EP1305Y, Etobicoke, ON, Canada) at a concentration of 1:300 at 4 °C overnight. After washing in PBS, biotinylated anti-rabbit IgG incubations were carried out. Immunoreactivity was revealed by incubation in DAB substrate (Vector labs, Burlington, ON, Canada) for 3 min. Slides were counterstained in Mayer's hematoxylin. MSI2 immunohistochemical staining was evaluated using light microscopy. Cases were blindly scored by a pathologist using the Allred Scoring system as described[37]. Available clinical data was used to compare MSI2 expression with tumour grade, stage and receptor status. Pearson's correlation (point biserial correlation for binary variables) was used to assess association at a 5% significance level.

**SPAG5 TMA**. Anti-SPAG5 rabbit antibody (HPA022008) was purchased from Sigma-Aldrich (St. Louis, MO, USA). FFPE TMA blocks of 234 UHN invasive breast cancers were stained with the antibody using the manufacturer's protocol. Scoring of immunohistochemical staining was performed using a semi-quantitative scoring system. TMAs were analysed in a blinded manner by a pathologist using a four-point semi-quantitative scale (0, 1+, 2+, 3+) for intensity. The intensity of staining was scored as 3+(strong), 2+(intermediate), 1+(weak) and 0 (no staining). Cases that were stained for SPAG5 showed homogeneous staining of all cancer cells within the core; the percentage positivity was 100%. Tissues showed moderate cytoplasmic positivity. As with MSI2, Pearson's correlation (point biserial correlation for binary variables) was used to assess association at a 5% significance level.

**Data availability**. Gene expression and aCGH data has been deposited by M.A. and S.J.D. in the National Center for Biotechnology Information's (NCBI's) Gene Expression Omnibus (GEO) and are accessible through GEO Series accession number GSE72653 (SuperSeries consisting of patient-matched gene expression [GSE72644] and aCGH [GSE72652] data). Correspondence and requests for materials should be addressed to S.J.D.

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

## Acknowledgements

We thank Dr. Kenneth Aldape for reading the manuscript and his helpful comments. This work is supported by grants from the Canadian Breast Cancer Foundation, Ontario, the Ontario Cancer Research Network, the Canadian Institutes for Health Research (MOP-119469) and the Princess Margaret Cancer Centre Weekend Walkers Fund (to S.J.D.). M.A. was supported in part by an Undergraduate Life Sciences Award from the Department of Laboratory Medicine and Pathobiology at the University of Toronto. N.K. and D.T.-T. received Fellowship funding from the Canadian Breast Cancer Foundation, Ontario. D.T.-T., J.M. and V.I. were supported by the Terry Fox Foundation Strategic Health Research Training Program in Cancer Research at CIHR—STP-53912. This study was conducted with the support of the Ontario Institute for Cancer Research to P.C.B. through funding provided by the Government of Ontario. P.C.B. was supported by a Terry Fox Research Institute New Investigator Award. This work also was supported in part by the Ontario Ministry of Health and Long Term Care. The views expressed in this paper do not necessarily reflect those of the Ontario Ministry of Health and Long Term Care.

## Author contributions

S.J.D. designed the study and supervised the research. Moustafa. A. designed the bioinformatic analysis. Ductoscopy samples were obtained and microdissected by D.T.-T. and V.I. with the help of W.L.L., A.M.E. and D.M. J.M. scored the TMAs. N.A. and K.W. assisted with sample preparation. N.K., R.N., J.Y.Y.K. and T.R.C., performed the IF and all cell line experiments. J.P.Y.L. performed and analysed the luciferase assays. S.J.D., N.A.M. and B.J.Y. assembled the TMAs. Moustafa. A., Mohamed. A., D.-Y.W., N.S.F. and P.C.B. consolidated, prepared and analysed all bioinformatics data. Moustafa. A. wrote the paper with assistance from S.J.D., P.C.B., N.K. and R.N. All authors discussed the results and commented on the manuscript.

## Additional information

**Competing interests:** The authors declare no competing financial interests.

