## [Peer Review File · Nature Communications]

Reviewers' comments:

Reviewer #1 (Remarks to the Author):

DESCRIPTION. This manuscript describes an analysis of genome copy number, mRNA expression and miRNA expression in biopsies taken from breast tumors, and ductal regions located proximal and distal to the tumor. The study reports several findings:

1. CNAs in ductal regions proximal to the tumor but none were recurrent. However, CNAs were found in ducts distal to the tumor that are likely CNA polymorphisms. Most were found throughout the individual patient tissue but some were found only in some samples suggesting that they arose during breast development.
2. Gene expression changes in ductal structures proximal and distal to the tumor were noted but again, none were recurrent.
3. Gene expression modules seem to be regulated by miRNAs whose expression levels are related to proximity to the tumor and may be early breast cancer determinants. Expression of the Wnt-related gene MSI2 is in the module most strongly associated with proximity to the tumor.

CRITIQUE. This report describes spatially defined 'omic changes in breast cancers that will be of substantial interest to the breast cancer research community. To my knowledge, it is this first study of this type in breast cancer. The data will be of interest to the general community and it is good to see that they have been deposited. The identification of miRNA driven changes in gene expression modules involving Wnt associated with proximity to the tumor is of particular interest. However, the manuscript might be improved in several areas prior to publication.

1. The gene expression modules were identified by assessment of gene co-expression without determining whether the co-expression patterns coordinately resulted in pathway activation. Use of an algorithm like PARADIGM (PMID: 20529912) would allow this to be assessed. The results would be more compelling if coordinate activation was observed.
2. It is surprising that differential expression of genes associated with early breast cancer initiation reported in other histologically defined studies were not detected in the present study (e.g. see PMID:15050918).
3. The authors assert that "if a module is involved in tumor initiation and development ...it must be enriched for univariately prognostic genes". It is not clear that this must be true. Tumor progression is a multistep process the end result of which may be lethal metastasis. But not all early lesions tumor progress to metastasis. Happily most do not. It seems likely that events subsequent to initiation strongly influence metastasis and outcome. Thus it seems quite possible that early events may not be prognostic.
4. It would be interesting to know the histological composition (epithelium vs stroma) of the biopsies if that information is available.
5. Regions of CNA are identified in the regions proximal and distal to the tumor. It would be helpful to have more information on the gene contents of these. Also, the fact that some are not present in all biopsies suggests that the event arose during mammary gland development. This deserves comment.
6. The associate of MSI2 expression with subtype and outcome is not particularly informative given that hundreds of subtype specific genes show such associations. This might be deleted from the manuscript.
7. It would be good to comment in the body of the paper on what overexpression of MSI2 does to the overall activity of the gene expression module with which it is associated.
8. The selection of SPAG5 as a representative gene in Module I seems completely ad hoc. A better justification of why this gene is of interest is needed.
8. The miRNA modules are not adequately described and the description of how these relate to the expression levels of module E genes is very superficial. The part of the manuscript needs to be strengthened and to be tested experimentally.

Reviewer #2 (Remarks to the Author):

This manuscript describes an interesting approach to study the earliest genomic and transcriptomic changes occurring in the breast leading to breast carcinoma. Tissue samples are obtained from several sites along the duct leading up to the tumor. Gene expression and array comparative hybridization microarrays are used to obtain comprehensive profiles. Several bioinformatic analyses are used to generate gene modules and identify genes and pathways that may be involved in early breast tumorigenesis; some of which are prognostic in an external dataset. Specifically they highlight two novel genes MSI2 and SPAG5. Functional assays are used to further study a candidate WNT pathway gene (MSI2) as well as tissue microarrays to investigate expression of MSI2 and SPAG5 across a cohort of breast tumors. Using miRNA expression data from a separate set of DCIS samples they hypothesize that miRNA may be responsible for the earliest changes taking place in breast carcinogenesis. It is particularly valuable that they are comparing normal and tumor epithelial tissue from the same patient, avoiding the dilemma of contrasting gene expression patterns between different patients, as used by most other studies. The idea and approach is interesting but the manuscript suffers from some limitations which are detailed below. The methods are sound but would benefit from some additional clarification. Quality of data is good Presentation can be improved, see specific comments.

Specific comments:

Line 99. Introduction. The sentence "More generally, the mRNA-miRNA coexpression..." indicate more than is actually shown in this paper on the mechanistic basis of early changes occurring in carcinogenesis. Besides, it is more appropriate to speculate in the Discussion than in Introduction.

Line 108. Results. State early in this section how large n is.

It is stated that the breast cancer cohort represent multiple molecular subtypes. What are these?

Line 123. The hypothesis of why there should be an enrichment of genomic changes in adjacent-to-tumor tissue compared with more distant tissue is not well explained. Have they also analyzed differences between adjacent to tumor tissue (T2) and tumor cells (T4)?

Is there a maximum length of the genomic intervals that they investigate?

Line 141. Numbers of aberrations are stated (305 of 378). Which analysis do these come from? In which table are these results shown? This should be clearly stated in this section.

Line 191. What is the size of each mRNA module? For module E, on line 213 it is stated 816 and in Table 1 it states 815. It would be very helpful with an overview of the modules with gene sizes. It could be added to Figure 2.

Line 204. It is stated that they deconvolute epithelial and stromal signatures without any explanation as to how this is achieved. After consulting the method section, it is referred to a method (ISOPure2) and a short description is provided. It is not clearly stated whether this separation of epithelial and stroma profiles is important for the prognostic significance of the identified genes. The reasoning for performing this analysis must be better explained.

Line 224. The statement on the corollary of this implication that other genes in the module MUST be involved in WNT signaling is not explained. Why – since the genes in the module are identified based on co-expression associations?

Line 242. MSI2 overexpression causes beta-catenin translocation. How? It has not been proven that this leads to expression of downstream Wnt targets (line 369 Discussion).

Line 246. Which molecular subtypes express MSI2?

Line 288. Were the same TMA used for SPAG5 and MSI2, this is not clear. 232-vs 234 cases ?

Line 292. This is an interesting analysis which should be further pursued. The result on mRNA-miRNA coexpression modules indicating that miRNAs in defined modules are the drivers of specific mRNA modules is hypothetical and should be further investigated and corroborated, as also indicated in the conclusions by the authors themselves.

Line 335. Discussion. What is meant by the sentence that most mutations occur after initiating events?

Line 369. In this section of Discussion they describe how miRNA-mRNA coexpression perturbations

are early aberrations in tumorigenesis and that these two types of expression modules are linked to specific signaling pathways and prognostic genes in the Metabric data. However, the miRNA data seem not to have been used to create the initial modules that were the basis for these analyses? The miRNA – mRNA coexpression modules are described later and for different reasons. This is quite confusing the way it is discussed.

Line 363. What are clinical covariates in vitro?

Line 372. It is stated that other studies have revealed amplification of the MSI2 genomic region which emphasizes its importance. The references given do not discuss MSI2 but other genes”, hence its candidate importance is an overstatement.

Line 374. Evidence of chromosomal instability with SPAG5, what does this mean?

Figures:

There are mistakes in the figure numbering; f ex lines 197-198; line 230, should be Figure S4b, and not S3a. Figure 3a is not referred to before 3b. Also, figure sub-numbers should be used in the main text when referring to specific panels (a, b, etc.)

Fig 1. The label “region” must be explained along with the other parameters below the dendrogram

Fig 3. The heatmap is poorly explained. In the legend MSI2 is mentioned but it is not shown in the figure. How is MSI2 relevant on this figure? It is not referred to Figure 3a in the main text. Fig 4. Same for 4d and SPAG5.

Fig 5. Is the MCF7 MSI2 control the same in c and e? If so, why do they show different migration and invasion counts? And is this difference significant?

Line 337. The authors emphasize that they are looking at spatial distance in their analyses, yet the title says "temporally" which is a bit misleading. The samples were obtained simultaneously and temporal patterns (in time) cannot easily be inferred from such data.

Other comments:

Define abbreviations when first used (WGCNA in abstract).

HUGO nomenclature should be followed consistently for genes and proteins in text and figures. US and UK English is both used (tumor/tumour).

Reviewer #3 (Remarks to the Author):

The authors obtained multi-omics data containing copy number, mRNA and miRNA expression profiles of breast tumors and surrounding epithelial tissues. they also performed bioinformatics analysis to obtain insights into earliest genomic events leading to tumorigenesis breast tumor. Although the experimental design of the data set might be unique, I think the bioinformatics analysis was not sufficiently refined to integrate multi-omics data and obtain biologically interesting insights; they simply applied a popular co-expression analysis, WGCNA, to mRNA and miRNA analysis while little analysis seemed to be done for copy number. As far as I read the paper, I cannot find any novel and solid findings that can be deduced uniquely from the multi-omic data set. From these reasons, I think that this work is not worth publication.

Reviewer 1 – Point 1: The gene expression modules were identified by assessment of gene co-expression without determining whether the co-expression patterns coordinately resulted in pathway activation. Use of an algorithm like PARADIGM (PMID: 20529912) would allow this to be assessed. The results would be more compelling if coordinate activation was observed.

Response: We've analyzed the combined aCGH and expression data using PARADIGM (both independently and through the ReactomeFI Cytoscape plugin). Unfortunately, the dataset doesn't appear well-powered enough to identify any activated pathways or perturbed molecules across nearly all samples.

Reviewer 1 – Point 2: It is surprising that differential expression of genes associated with early breast cancer initiation reported in other histologically defined studies were not detected in the present study (e.g. see PMID:15050918).

Response: The article referenced analyzes reduction mammoplasty specimens from five individuals; these are likely to be from patients with mammary hypertrophy and although histologically normal may not be normal at a molecular level.. However, considering the heterogeneity of breast cancer, it is likely that neither dataset is powered enough to make concrete claims regarding differential expression in adjacent-to-tumor epithelia. From our review of literature (see text), no such dataset currently exists. We've elected to replace our "classical" differential expression with a co-clustering approach; representing a more robust way of identifying sets of genes that are unregulated in any particular distance annotation (see lines 156-182, Methods and Figure 2 for details). We also show that the first principal component of the expression matrix largely accounts for proximity to tumor (lines 119-122 and Figure 1). This suggests that expression changes may be gradual along the tumor duct.

Reviewer 1 – Point 3: The authors assert that "if a module is involved in tumor initiation and development ...it must be enriched for univariately prognostic genes". It is not clear that this must be true. Tumor progression is a multistep process the end result of which may be lethal metastasis. But not all early lesions tumor progress to metastasis. Happily most do not. It seems likely that events subsequent to initiation strongly influence metastasis and outcome. Thus it seems quite possible that early events may not be prognostic.

Response: We've removed that statement and our prognostic analysis.

Reviewer 1 – Point 4: It would be interesting to know the histological composition (epithelium vs stroma) of the biopsies if that information is available.

Response: These were microdissected samples composed of > 90% epithelial cells (we have clarified the histological composition in line 108).

Reviewer 1 – Point 5: Regions of CNA are identified in the regions proximal and distal to the tumor. It would be helpful to have more information on the gene contents of these. Also, the fact that some are not present in all biopsies suggests that the event arose during mammary gland development. This deserves comment.

Response: Supplementary Table 2 contains the gene contents of the regions of CNA. We have added a comment in the manuscript regarding this observation (lines 129-132).

Reviewer 1 – Point 6: The associate of MSI2 expression with subtype and outcome is not particularly informative given that hundreds of subtype specific genes show such associations. This might be deleted from the manuscript.

Response: We've removed discussion of the association of MSI2 with subtype from the manuscript. We've elected to keep the results as reference, but do not address them in detail.

Reviewer 1 – Point 7: It would be good to comment in the body of the paper on what overexpression of MSI2 does to the overall activity of the gene expression module with which it is associated.

Response: It is unclear what MSI2 does to the overall activity of the gene expression module. We know that MSI2 overexpression causes an activation of WNT signalling and that the module is enriched for genes associated with “formation of the beta-catenin:TCF transactivating complex” gene set. While this hints at a causal relationship (i.e. MSI2 driving the module), it would require more extensive MSI2-specific data. We briefly summarize this in lines 219-222.

Reviewer 1 – Point 8: The selection of SPAG5 as a representative gene in Module I seems completely ad hoc. A better justification of why this gene is of interest is needed.

Response: The selection was arbitrary; among highly correlated modules, we selected those with genes for which we had tissue microarray data (to supplement and partially validate our *in silico* analyses). This justification has been added to the manuscript (lines 186-188).

Reviewer 1 – Point 9: The miRNA modules are not adequately described and the description of how these relate to the expression levels of module E genes is very superficial. The part of the manuscript needs to be strengthened and to be tested experimentally.

Response: As we did not experimentally verify the relationship between the miRNA modules and transcriptomic perturbations, we've elected to remove the miRNA analysis. In lieu of that, we demonstrate that CNAs (i.e. genetic aberrations) are not sufficient to explain the observed expression perturbation with decreasing proximity to tumor (see lines 118-132 and Supplementary Table 2).

Reviewer 2 – Point 1: Line 99. Introduction. The sentence “More generally, the mRNA-miRNA coexpression...” indicate more than is actually shown in this paper on the mechanistic basis of early changes occurring in carcinogenesis. Besides, it is more appropriate to speculate in the Discussion than in Introduction.

Response: We have removed that statement.

Reviewer 2 – Point 2: Line 108. Results. State early in this section how large n is. It is stated that the breast cancer cohort represent multiple molecular subtypes. What are these?

Response: We've stated how large n is (n = 9 patients) in both the Introduction and Results. The multiple molecular subtypes were referenced in the previous figure 1 and have been added to the main text (line 114).

Reviewer 2 – Point 3: Line 123. The hypothesis of why there should be an enrichment of genomic changes in adjacent-to-tumor tissue compared with more distant tissue is not well explained. Have they also analyzed differences between adjacent to tumor tissue (T2) and

tumor cells (T4)? Is there a maximum length of the genomic intervals that they investigate?

Response: To address these questions, we show that after adjusting for baseline expression (i.e. using the epithelial sample from the contralateral duct) genomic changes cannot explain the increasing perturbation observed with proximity to the tumor (nor within the tumor itself); results are presented in lines 119 – 133.

Reviewer 2 – Point 4: Line 141. Numbers of aberrations are stated (305 of 378). Which analysis do these come from? In which table are these results shown? This should be clearly stated in this section.

Response: We have replaced this section with the modelling and variance analysis described in Point 3 (above); lines 119 – 133.

Reviewer 2 – Point 5: Line 191. What is the size of each mRNA module? For module E, on line 213 it is stated 816 and in Table 1 it states 815. It would be very helpful with an overview of the modules with gene sizes. It could f ex be added to Figure 2.

Response: The difference was addressed in the caption of the table: mapping between the different nomenclatures (e.g. entrez id or gene symbol) is not one-to-one, i.e. not bijective. Thus, on occasions, the gene count would differ. On a related note, we've opted to implement a biclustering algorithm (instead of WGCNA) to leverage our knowledge of sample distance-to-tumor to identify more relevant modules. The key results (such as enriched pathways) did not change; only slight changes in module definitions/size were noted.

Reviewer 2 – Point 6: Line 204. It is stated that they deconvolute epithelial and stromal signatures without any explanation as to how this is achieved. After consulting the method section, it is referred to a method (ISOpure2) and a short description is provided. It is not clearly stated whether this separation of epithelial and stroma profiles is important for the prognostic significance of the identified genes. The reasoning for performing this analysis must be better explained.

Response: At the suggestion of reviewer 1, we've removed the prognostic analysis section.

Reviewer 2 – Point 7: Line 224. The statement on the corollary of this implication that other genes in the module MUST be involved in WNT signaling is not explained. Why – since the genes in the module are identified based on co-expression associations?

Response: We've changed the phrasing of the sentence (line 194). Genes in coexpression modules tend to have similar function (i.e. guilt by association)².

Reviewer 2 – Point 8: Line 242. MSI2 overexpression causes beta-catenin translocation. How? It has not been proven that this leads to expression of downstream Wnt targets (line 369 Discussion).

Response: Although we observe clear evidence that MSI2 overexpression causes nuclear localization of beta-catenin in MCF7, the mechanism is unclear. However, we do observe significant action of TCF/LEF-1–dependent luciferase reporter construct in MCF7 (i.e. activation of Wnt signaling and likely expression of downstream Wnt targets).

Reviewer 2 – Point 9: Line 246. Which molecular subtypes express MSI2?

Response: MSI2 is positively correlated with ER and PR positive status.

Reviewer 2 – Point 10: Line 288. Were the same TMA used for SPAG5 and MSI2, this is not clear. 232-vs 234 cases?

Response: Yes, the same TMA was used. However, 2 cores were spoiled in the MSI2 TMA. This is mentioned in the Methods (lines 231-234).

Reviewer 2 – Point 11: Line 292. This is an interesting analysis which should be further pursued. The result on mRNA-miRNA coexpression modules indicating that miRNAs in defined modules are the drivers of specific mRNA modules is hypothetical and should be further investigated and corroborated, as also indicated in the conclusions by the authors themselves.

Response: At the suggestion of Reviewer 1 and Reviewer 3, we've removed the miRNA analysis. In lieu of that, we show that genetic aberrations are not sufficient to explain transcriptomic perturbations observed with proximity to tumor.

Reviewer 2 – Point 12: Line 335. Discussion. What is meant by the sentence that most mutations occur after initiating events?

Response: Stephens et al. ¹ suggested that most genomic aberrations occur after the initiating event (i.e. the 'seed'). In other words, genomic alterations are not necessary for tumor initiation.

Reviewer 2 – Point 13: Line 369. In this section of Discussion they describe how miRNA-mRNA coexpression perturbations are early aberrations in tumorigenesis and that these two types of expression modules are linked to specific signaling pathways and prognostic genes in the METABRIC data. However, the miRNA data seem not to have been used to create the initial modules that were the basis for these analyses? The miRNA – mRNA coexpression modules are described later and for different reasons. This is quite confusing the way it is discussed.

Response: At the suggestion of Reviewer 1 and Reviewer 3, we've removed the miRNA analysis.

Reviewer 2 – Point 14: Line 363. What are clinical covariates in vitro?

Response: At the suggestion of Reviewer 1, we had removed the METABRIC univariate prognostic analysis.

Reviewer 2 – Point 15: Line 372. It is stated that other studies have revealed amplification of the MSI2 genomic region which emphasizes its importance. The references given do not discuss MSI2 but other genes", hence its candidate importance is an overstatement.

Response: We've rephrased the sentence to de-emphasize MSI2's candidate importance (line 323).

Reviewer 2 – Point 16: Line 374. Evidence of chromosomal instability with SPAG5, what does this mean?

Response: We've removed that sentence; instead, electing to focus on SPAG5's correlation with tumor grade in the TMA.

Reviewer 2 – Point 16: There are mistakes in the figure numbering; for example lines 197-198; line 230, should be Figure S4b, and not S3a. Figure 3a is not referred to before 3b. Also, figure sub-numbers should be used in the main text when referring to specific panels (a, b, etc.)

Response: These figures and figure references have been removed.

Reviewer 2 – Point 17: Fig 1. The label “region” must be explained along with the other parameters below the dendrogram

Response: The figure has been replaced.

Reviewer 2 – Point 18: Fig 3. The heatmap is poorly explained. In the legend MSI2 is mentioned but it is not shown in the figure. How is MSI2 relevant on this figure? It is not referred to Figure 3a in the main text. Fig 4. Same for 4d and SPAG5.

Response: The figure has been removed.

Reviewer 2 – Point 19: Fig 5. Is the MCF7 MSI2 control the same in c and e? If so, why do they show different migration and invasion counts? And is this difference significant?

Response: No, the controls are not the same. Figure 5c represents transient transfections experiments. Figure 5e represents experiments with permanent sh clones. This was referenced in the captions.

Reviewer 2 – Point 20: Line 337. The authors emphasize that they are looking at spatial distance in their analyses, yet the title says "temporally" which is a bit misleading. The samples were obtained simultaneously and temporal patterns (in time) cannot easily be inferred from such data.

Response: We've updated the title and removed the word “temporally”.

Reviewer 2 – Point 21: Define abbreviations when first used (WGCNA in abstract).

Response: In lieu of WGCNA, we apply a co-clustering algorithm to leverage the sample annotations (i.e. distance to tumor) to identify co-expression modules. We've updated the text accordingly.

Reviewer 2 – Point 22: HUGO nomenclature should be followed consistently for genes and proteins in text and figures. US and UK English is both used (tumor/tumour).

Response: We have corrected the nomenclature and spelling.

Reviewer 3 – Point 1: The authors obtained multi-omics data containing copy number, mRNA and miRNA expression profiles of breast tumors and surrounding epithelial tissues. they also performed bioinformatics analysis to obtain insights into earliest genomic events leading to tumorigenesis breast tumor. Although the experimental design of the data set might be unique, I think the bioinformatics analysis was not sufficiently refined to integrate multi-omics data and

obtain biologically interesting insights; they simply applied a popular co-expression analysis, WGCNA, to mRNA and miRNA analysis while little analysis seemed to be done for copy number. As far as I read the paper, I cannot find any novel and solid findings that can be deduced uniquely from the multi-omic data set. From these reasons, I think that this work is not worth publication.

Response: We've attempted to address the concerns of Reviewer 3 by completing new analyses that leverage the unique dataset design (highlighted in the introduction of this reply). Briefly, we showed that (after adjusting for baseline expression) almost none of the variance in transcriptomic perturbations can be explained by copy number aberrations. Furthermore, there is no dependence on distance to tumor. We also showed that the first principal component of the expression matrix corresponds to proximity to tumor and highlight some of the genes/features that contribute to this component. Subsequently, we demonstrate that a multi-label classifier can identify from where the epithelial samples were obtained relative to the tumor (with a cross-validated AUC = 0.74). We discuss the features used in this classifier and more importantly the implications: direct evidence of field cancerization in expression data. Furthermore, to leverage our knowledge of sample annotations (i.e. the distance to tumor), we used a coclustering (equivalently, biclustering) algorithm to define coexpression modules integrated with a hierarchical-cluster driven feature agglomeration for dimensionality reduction. Unlike with WGCNA, we select the number of modules that subdivided the sample into classes that closely match the original distance annotations. We believe this logical approach addresses the main concerns of Reviewer 3: highlighting novel and solid findings that are only possible because of the uniqueness of the dataset design.

REFERENCES

- 1 Stephens, P. J. *et al.* The landscape of cancer genes and mutational processes in breast cancer. *Nature* **486**, 400-404, doi:10.1038/nature11017 (2012).
- 2 Gaiteri, C., Ding, Y., French, B., Tseng, G. C. & Sibille, E. Beyond modules and hubs: the potential of gene coexpression networks for investigating molecular mechanisms of complex brain disorders. *Genes, Brain and Behavior* **13**, 13-24 (2014).

Reviewers' comments:

Reviewer #1 (Remarks to the Author):

DESCRIPTION. This revised manuscript assesses gene expression and copy number abnormalities (CNAs) in epithelial samples taken proximal and distal to tumor in 9 patients. The study shows that gene expression changes are greater proximal to the tumor than distal and that the gene expression changes are not explained by CNAs. They also assess two genes from gene expression modules that are differentially expressed between distal and proximal samples with the idea that these are somehow implicated in tumorigenesis. Specifically they show that manipulation of the expression of MSI2, member of the Wnt pathway, increases beta catenin activity and invasive behavior in MCF7 cells and with grade, stage, and receptor status in human tumors. They also show that SPAG5 is positively associated with tumor grade.

CRITIQUE. The revised manuscript addresses several issues raised in the last review. The most important finding in the paper is the association of gene expression modules whose expression levels change with proximity to the tumor and that the changes are not explained by genome copy number changes and are the same between patients. This is an important observation and worth publishing. However, three issues remain to be addressed. Specifically.

1. The authors suggest that differential expression of genes implicates these genes as causally related to tumorigenesis. It is equally plausible that these genes are differentially expressed because they are responding to signals from the tumor. This needs to be addressed.
2. The selection of MSI2 and SPAG5 as genes causally related to tumorigenesis remains completely ad hoc. There might be hundreds of other genes that show similar associations. Thus, nothing seems to make them special. A stronger driving motivation is needed to implicate these genes as causally related to breast tumorigenesis. The authors also might test whether expression changes in response to paracrine signaling from the tumor.
3. Inspection of the genes expression modules in supplemental Table 6 shows a number involved in epigenomic function. It is reasonably well accepted in the breast cancer community that the earliest events in breast tumorigenesis are epigenomic in nature. The authors might strengthen their argument that the gene expression changes are causally related to tumorigenesis by focusing on these epigenomic modules.

Reviewer #2 (Remarks to the Author):

Comments to the revised manuscript of Abdalla et al., Mapping genomic and transcriptomic alterations spatially in epithelial cells adjacent to human breast carcinoma.

The revised manuscript is much improved and appears more clear and focused. The methods are more clearly described, the language is significantly improved, figures are better and some of the statements that were too speculative have been rephrased. The whole part on miRNA analysis is now taken out, which was wise as it appeared immature. However, I hope the authors will continue these analyses as the mRNA-miRNA co-expressions are potentially very interesting in early tumorigenesis.

Minor comments:

1. Molecular subtypes-how were they determined? If by standard clinical measurements, state the subtypes by ER/PR/HER2.
2. The phrase "temporal analysis" is still included in the "first sentence of the Results section. Should be taken out.
3. It would be helpful if the individual figure panels are referred to throughout the text.
4. One conclusion is that CNA cannot explain the gene expression variations that gradually

increase towards the tumor, and that these effects are benign. In this context, I do not understand the use of "benign". Effects are limited which is sufficient to state.

5. In the section on MSI2 line 187, it is stated that the authors sought to explore the biological/clinical relevance of the mRNA module in questions. This is at best, biological exploration and not clinical.

6. Legend to Supplementary Figure 1 has a mistake: SPAG6

7. In the Bioinformatic analysis section, line 384, it says "microarray and aCGH data". Be more specific when describing gene expression data as both data types are from microarrays.

8. In Discussion second section line 276, it is stated increased distance but it should be the opposite.

9. When discussing SPAG5, be more specific that it is the expression of SPAG5 that positively correlates with grade and that expression of SPAG5 might be an early event.

10. Explain the clinico - pathological features displayed in Supplementary Table 1 in the legend.

Reviewer #3 (Remarks to the Author):

I appreciated that the authors extensively revised the manuscript. According to my comments, they found a correlation between spatial information and expression profiles. They insisted that this correlation is evidence of field cancerization. However, I cannot understand this logic. In the introduction, the authors said that "Others have suggested that presence of contaminating tumor cells beyond the invasive tumor margin, rather than field cancerization, may be the cause of local recurrence." (an explanation about field cancerization should be put before this sentence.) This data seems to support this hypothesis. And also it is possible this data reflects the fact that tumor cells produce secretory factors that change gene expression in surrounding normal cells. For acceptance of their insistence, it is necessary that the authors exclude these possibilities.

Thank you very much for your insightful comments and providing us with the opportunity to further revise and improve our manuscript, titled “**Mapping genomic and transcriptomic alterations spatially in epithelial cells adjacent to human breast carcinoma**”. The manuscript has been revised as recommended by the reviewers and in particular, addresses limitations about inferring causality from the observed transcriptomic “field of perturbation” surrounding the tumor.

Reviewer 1 – Point 1: The authors suggest that differential expression of genes implicates these genes as causally related to tumorigenesis. It is equally plausible that these genes are differentially expressed because they are responding to signals from the tumor. This needs to be addressed.

We consolidated mutation data from 8 different breast cancer datasets, including The Cancer Genome Atlas (TCGA)^{1,2}, METABRIC³, and datasets from the Broad⁴, Sanger⁵, and British Columbia Cancer Research Centre⁶ (curated by the cBIO Cancer Genomics Portal⁷). We show that genes in coexpression modules that correlate with increasing proximity to the primary tumor are more likely to be mutated in cancer patients, compared to background (Chi-square test; Yates p-value <0.0001; odds ratio = 1.44). Results are presented in lines 184 to 201 of the manuscript.

Reviewer 1 – Point 2: The selection of MSI2 and SPAG5 as genes causally related to tumorigenesis remains completely ad hoc. There might be hundreds of other genes that show similar associations. Thus, nothing seems to make them special. A stronger driving motivation is needed to implicate these genes as causally related to breast tumorigenesis. The authors also might test whether expression changes in response to paracrine signaling from the tumor.

Our selection of MSI2 and SPAG5 was not based on genes causally related to tumorigenesis, but based on whether we had tissue microarray data available. We were primarily interested in validating the observed transcriptomic perturbation with another data type and in a larger cohort. Our analyses from Point 1 (aforementioned) suggest

that the module/transcriptomic perturbation is likely upstream of tumorigenesis (rather than in response to paracrine signaling from the tumor).

Reviewer 1 – Point 3: Inspection of the gene expression modules in supplemental Table 6 shows a number involved in epigenomic function. It is reasonably well accepted in the breast cancer community that the earliest events in breast tumorigenesis are epigenomic in nature. The authors might strengthen their argument that the gene expression changes are causally related to tumorigenesis by focusing on these epigenomic modules.

Unfortunately, the enrichment in Supplementary Table 6 is limited to one module; thus we cannot extend the conclusions to the larger “field” transcriptomic perturbations we observe. However, as highlighted in point 1 (aforementioned), we are able to implicate this dysregulation as a consequence of field cancerization through mutation data analysis.

Reviewer 2 – Point 1: Molecular subtypes-how were they determined? If by standard clinical measurements, state the subtypes by ER/PR/HER2.

They were determined by ER/PR and HER2 status. However, as the numbers are so small we can't conclude anything about a particular subtype. We've removed reference to the molecular subtypes.

Reviewer 2 – Point 2: The phrase “temporal analysis” is still included in the first sentence of the Results section. Should be taken out.

We have removed the phrase (from line 97).

Reviewer 2 – Point 3: It would be helpful if the individual figure panels are referred to throughout the text.

We have updated the text to reference individual figure panels.

Reviewer 2 – Point 4: One conclusion is that CNA cannot explain the gene expression variations that gradually increase towards the tumor, and that these effects are benign. In this context, I do not understand the use of “benign”. Effects are limited which is sufficient to state.

We have removed the word benign (lines 132-133).

Reviewer 2 – Point 5: In the section on MSI2 line 187, it is stated that the authors sought to explore the biological/clinical relevance of the mRNA module in questions. This is at best, biological exploration and not clinical.

We have removed the word clinical (line 187).

Reviewer 2 – Point 6: Legend to Supplementary Figure 1 has a mistake: SPAG6

The legend to Supplementary Figure 1 has been corrected.

Reviewer 2 – Point 7: In the Bioinformatic analysis section, line 384, it says “microarray and

aCGH data". Be more specific when describing gene expression data as both data types are from microarrays.

We have updated the text (line 384) to say "mRNA abundance and aCGH data".

Reviewer 2 – Point 8: In Discussion second section line 276, it is stated increased distance but it should be the opposite.

We have corrected the sentence (line 276) to state "with proximity to tumor".

Reviewer 2 – Point 9: When discussing SPAG5, be more specific that it is the expression of SPAG5 that positively correlates with grade and that expression of SPAG5 might be an early event.

We have emphasized that the protein abundance of SPAG5 positively correlates with grade (line 240). At the suggestion of Reviewers 1 and 3, we limited our conclusions on inferring causality/temporal inferences based on these results.

Reviewer 2 – Point 10: Explain the clinico - pathological features displayed in Supplementary Table 1 in the legend.

We have added an explanation of the clinico-pathological features in the legend of Supplementary Table 1.

Reviewer 3 – Point 1: In the introduction, the authors said that "Others have suggested that presence of contaminating tumor cells beyond the invasive tumor margin, rather than field cancerization, may be the cause of local recurrence." (an explanation about field cancerization should be put before this sentence.) This data seems to support this hypothesis.

We've added an explanation of field cancerization in the introduction (lines 80-81). We can exclude the possibility of contaminating tumor cells as all specimens were manually microdissected and obtained under direct vision (making the likelihood of tumour cell contamination very low); discussed in lines 189-191 and elsewhere. This is a major strength of our approach.

Reviewer 3 – Point 2: And also it is possible this data reflects the fact that tumor cells produce secretory factors that change gene expression in surrounding normal cells. For acceptance of their insistence, it is necessary that the authors exclude these possibilities.

We can exclude paracrine influence of the primary tumor by analyzing mutation data from 8 different breast cancer datasets (including the METABRIC and TCGA cohorts). In particular, we note that genes in coexpression modules with significant correlation to proximity to tumor are more likely to be mutated in cancer patients, compared to background (Chi-square test; Yates p-value <0.0001; odds ratio = 1.44). Results are presented in lines 184 to 204 of the manuscript. This genetic evidence suggests the transcriptomic dysregulation are independent of paracrine influence from the tumor.

REFERENCES

- 1 Ciriello, G. *et al.* Comprehensive molecular portraits of invasive lobular breast cancer. *Cell* **163**, 506-519 (2015).
- 2 Network, C. G. A. Comprehensive molecular portraits of human breast tumors. *Nature* **490**, 61 (2012).
- 3 Pereira, B. *et al.* The somatic mutation profiles of 2,433 breast cancers refines their genomic and transcriptomic landscapes. *Nature communications* **7** (2016).
- 4 Banerji, S. *et al.* Sequence analysis of mutations and translocations across breast cancer subtypes. *Nature* **486**, 405-409 (2012).
- 5 Stephens, P. J. *et al.* The landscape of cancer genes and mutational processes in breast cancer. *Nature* **486**, 400-404 (2012).
- 6 Shah, S. P. *et al.* The clonal and mutational evolution spectrum of primary triple-negative breast cancers. *Nature* **486**, 395-399 (2012).
- 7 Cerami, E. *et al.* (AACR, 2012).

Reviewers' comments:

Reviewer #1 (Remarks to the Author):

This revised manuscript explores genomic and transcriptional changes in regions of apparently normal tissue that are distal and proximal to tumor lesions in 8 breast cancer patients. The data set will be of high interest to the breast cancer research community. The revised manuscript is more clearly written and much improved. However, there remains one issue that is not as convincing as it should be for a publication in Nature Communications. That is, the authors' assertion that their data supports field cancerization as the mechanism driving the association between gene expression and proximity to the tumor (as opposed to paracrine signaling).

They argue in the revised manuscript that a field cancerization mechanism would lead to a larger number of mutations in proximity association module genes. They show that the proximity association module genes were more frequently mutated in cancers than those in modules not associated with proximity to the cancer lesion in several large data sets. Unfortunately, this is not completely convincing. It could be that paracrine effects predispose affected cells to increased mutation. I suspect the authors are correct in their assertion but the logic presented in the paper is not compelling. Reference to publications that have explored field cancerization in normal tissue might strengthen their case (e.g. PMID: 18006786, 15753376). It would be good to comment specifically on INK4a and COX-2 expression in the proximity association modules.

Reviewer #2 (Remarks to the Author):

In the most recently revised manuscript the authors have addressed the comments I had from the previous round and the manuscript appears more solid and mature. The authors seem to have addressed comments from all reviewers and conclude that their gene expression data provide strong evidence for field cancerization. They have also included a new analysis in the Results section of the revised manuscript to further support their conclusions. This is an interesting hypothesis that is surrounded by some controversy and which should be interesting to follow up in later studies.

Specific comments:

- Use mutations instead of genetic variants on first page of Introduction. Genetic variants lead the reader to think these are germline. What are the related traits that these mutations are associated to (aside from breast cancer)?
- Remove the sentence on which subtypes are included in Supplementary Table 1 on first page of Results.
- I think it is a bold statement that the authors can exclude the possibility of contaminating tumor cells because they were manually microdissecting cells. Likelihood is low yes.
- Still not all genes/proteins follow nomenclature (Msi2)
- I find the last sentence of the Discussion superfluous.

Reviewer #3 (Remarks to the Author):

In response to Reviewer 3 – Point 1, the authors say that “We can exclude the possibility of contaminating tumor cells as all specimens were manually microdissected and obtained under direct vision (making the likelihood of tumor cell contamination very low)” on the other hand, the response to Reviewer 3 – Point 2 says that “This genetic evidence suggests the transcriptomic dysregulation are independent of paracrine influence from the tumor.”, which means they observed transcriptomic dysregulation in tumor cells. The authors examined normal cells or tumor cells? These two claims are contradictory and I cannot understand the rebuttal letter.

Reviewer 1 –

“This revised manuscript explores genomic and transcriptional changes in regions of apparently normal tissue that are distal and proximal to tumour lesions in 8 breast cancer patients. The data set will be of high interest to the breast cancer research community. The revised manuscript is more clearly written and much improved. However, there remains one issue that is not as convincing as it should be for a publication in Nature Communications. That is, the authors

assertion that their data supports field cancerisation as the mechanism drives the association between gene expression and proximity to the tumour (as opposed to paracrine signaling).

They argue in the revised manuscript that a field cancerisation mechanism would lead to a larger number of mutations in proximity association module genes. They show that the proximity association module genes were more frequently mutated in cancers than those in modules not associated with proximity to the cancer lesion in several large data sets. Unfortunately, this is not completely convincing. It could be that paracrine effects predispose affected cells to increased mutation. I suspect the authors are correct in their assertion but the logic presented in the paper is not compelling. Reference to publications that have explored field cancerisation in normal tissue might strengthen their case (e.g. PMID: 18006786, 15753376). It would be good to comment specifically on INK4a and COX-2 expression in the proximity association modules”

Reviewer 1 - Point 1: We thank the reviewer for the opportunity to improve this aspect of the paper. We believe the possibility of paracrine effects to be unlikely for several reasons.

1. We observed an increase in somatic mutations in gene sets that significantly correlate with proximity to tumour (*i.e.* there is an enrichment of mutations in only select genes compared to background). If paracrine effects predispose cells to increased mutations, all genes should be affected equally.
2. Also, if a paracrine effect is at play it is unlikely that it will extend so far beyond the tumour as our samples were taken and also that it has a similar effect regardless of the adjacent tissue type (fibrous, adipose, epithelial, etc). However, we cannot exclude a paracrine effect passing through the epithelium only.
3. In the current version, we have demonstrated that genes that correlate with proximity to tumour are enriched for somatic mutations in tumours (*i.e.* they are more likely to be mutated). This suggests a functional role for many of these genes in tumour development and progression. Furthermore, as aforementioned, this increased mutation rate among those genes is unlikely to be a consequence of paracrine influence (an increased mutation rate should affect most genes equally).
4. In addition, an article published this week in Nature ("Recurrent and functional regulatory mutations in breast cancer") is very relevant to our manuscript. We have updated the discussion to include the results of this article.

We have added lines **315-323** and **343-393** to the Discussion to include these points:

Lines 315-323:

As we discuss below, we showed that transcriptomic alterations in spatially-informative coexpression modules are likely **more important than the** paracrine influence of the tumour and this perturbation may precede tumourigenesis. **Furthermore, the enrichment of somatic mutations in genes that correlate with proximity to tumour suggests a non-paracrine mechanism underlying the dysregulation. If paracrine effects predispose cells to increased mutations, all genes should be affected equally (*i.e.* no enrichment). Furthermore, if a paracrine**

effect is at play, it is also unlikely that it will extend so far beyond the tumour as our samples were taken and that it would produce a similar effect regardless of the adjacent tissue type (fibrous, adipose, epithelial, etc).

Lines 339-351:

Subsequently, using spectral co-clustering, we identified biclusters (*i.e.* coexpressors) with perturbations that may be among the first molecular alterations shared by breast cancer patients, preceding common aberrations detected by aCGH. Genes in these biclusters were *a priori* selected as informative for proximity-to-tumour label. We demonstrate that the genes in modules with significant correlation to proximity to tumour were more likely to be mutated in breast cancer patients (compared to background genes in non-significant modules), suggesting that this perturbation is a consequence of field cancerisation (rather than paracrine influence from the primary tumour). As these genes are enriched for somatic mutations in tumours (*i.e.* they are more likely to be mutated), this further suggests a functional role for many of these genes in tumour development and progression. Furthermore, in an analysis by Rheinbay *et al.*³⁰, deep sequencing of 360 primary breast cancers revealed six promoters (TBC1D12, LEPROTL1, ZNF143, RMRP, ALDOA, and FOXA1) that contained single-site mutational hotspots (three or more mutations at a single site). Three genes (ZNF143, ALDOA, and LEPROTL1) were among the top 30% of genes that are univariately informative of the proximity-to-tumour label. Two genes (TBC1D12 and RMRP) were not on the expression platform; FOXA1 was not informative of the spatial label for the samples and was the only gene (among the 4 present on the platform) with the mutational hotspot not located in or adjacent to the 5' untranslated region. Two genes (ALDOA in module 5 and LEPROTL1 in module 2) also belonged to modules that significantly and positively correlated with proximity to tumour. Thus, non-coding single-nucleotide mutations may be responsible, in part, for the observed field cancerisation.

Reviewer 1 – Point 2: We thank the reviewer for these references (PMID: 18006786, 15753376) and have added them as suggested (references #8 and #12). We comment on the role of MAPK14 (p38) and CDKN2A (INK4a) on lines 365-371 and lines 387-393. Unfortunately COX2 is not present on the platform (by any name).

Reviewer 2 –

“In the most recently revised manuscript the authors have addressed the comments I had from the previous round and the manuscript appears more solid and mature. The authors seem to have addressed comments from all reviewers and conclude that their gene expression data provide strong evidence for field cancerisation. They have also included a new analysis in Result section of the revised manuscript to further support their conclusions. This is an interesting hypothesis that is surrounded by some controversy and which should be interesting to follow up in later studies.

Specific comments:

- Use mutations instead of genetic variants on first page of Introduction. Genetic variants lead the reader to think these are germline. What are the related traits that these mutations are associated to (aside from breast cancer)
- Remove the sentence on which subtypes are included in Supplementary Table 1 on first page of Results.

- I think it is a bold statement that the authors can exclude the possibility of contaminating tumour cells because they were manually microdissecting cells. Likelihood is low yes.
- Still not all genes/proteins follow nomenclature (Msi2)
- I find the last sentence of the Discussion superfluous.“

Reviewer 2 – Point 1: Thank you, we have updated the Introduction accordingly (line 90-92; changes are in red):

... and enrichment of somatic mutations commonly present in breast cancer suggests these perturbations were not a consequence of paracrine influence from the adjacent tumour.

We looked at whether or not a mutation was present in the genes (as a running total), but did not collect any information on the mutations themselves. The pathway analysis reveals roles for genes that are mutated and by extension, the possible impact of these mutations.

Reviewer 2 – Point 2: We have removed the sentence on subtypes.

Reviewer 2 – Point 3: We have updated the sentence (lines 195-196; changes in red):

As all specimens were manually microdissected by a pathologist and obtained under direct vision, the likelihood of tumour cell contamination was low.

Reviewer 2 – Point 4: We have updated all gene/protein names to follow nomenclature.

Reviewer 2 – Point 5: We have removed the last sentence of the Discussion.

Reviewer 3 –

“In response to Reviewer 3 – Point 1, the authors says that “We can exclude the possibility of contaminating tumour cells as all specimens were manually microdissected and obtained under direct vision (making the likelihood of tumour cell contamination very low)” on the other hand, the response to Reviewer 3 – Point 2 says that “This genetic evidence suggests the transcriptomic dysregulation are independent of paracrine influence from the tumour.”, which means they observed transcriptomic dysregulation in tumour cells. the authors examined normal cells or tumour cells? These two claims are contradictory and I cannot understand the rebuttal letter.”

We thank the reviewer for highlighting a lack of clarity.

We examined both normal and tumour cells. We collected samples from the duct leading to the tumour and a control sample from an opposite duct. All samples were needle microdissected. This reduces the likelihood of tumour or stromal contamination in the “normal” epithelial samples (*i.e.* O1, D1, and most of D2). An ASCII figure (resembling Figure 1 from the manuscript) showing the samples collected and analysed (O1, D1, D2 and Tumour) is shown below:

-----O1----- NIPPLE ----D1-----D2-----T(umor)

We have improved Figure 1 to show the samples analysed and renamed the samples from the duct leading to the tumour as D1 and D2 to emphasise that these are Ductal epithelial samples. Also, we have added mention of the analysis of the tumour to the first paragraph of the Discussion:

Lines 298-301:

By obtaining and profiling morphologically normal epithelial samples at various distances from the tumour (and a sample from the tumour itself), we were able to draft a spatial map of the genomic events and transcriptomic alterations that occur along the mammary duct (leading up to and including the tumour).

Please see response to Reviewer #1 for a discussion of the influence of a paracrine effect.

Thank you again for your constructive comments.
Yours sincerely,

Susan

REVIEWERS' COMMENTS:

Reviewer #1 (Remarks to the Author):

This revised manuscript explores genomic and transcriptional changes in regions of apparently normal tissue that are distal and proximal to tumour lesions in 8 breast cancer patients. The revised manuscript is clearly written and much improved. Specifically, the authors have made a compelling case for field cancerisation as the mechanism drives the association between gene expression and proximity to the tumour (as opposed to paracrine signaling). They show that the proximity association module genes were more frequently mutated in cancers than those in modules not associated with proximity to the cancer lesion in several large data sets. They also provide functional evidence suggesting how some of the deregulated genes can contribute to carcinogenesis. The manuscript and the associated data will be of high interest to the breast cancer research community.

Reviewer #2 (Remarks to the Author):

In the response to reviewer 2 the authors have addressed and satisfactorily answered questions and comments. The manuscript appears much improved and a bit more cautious in the conclusions of field cancerization. I think the manuscript deserves to be published and will represent a very valuable data set to continue to investigate.

Reviewer #3 (Remarks to the Author):

Although the manuscript has been improved to some degree, I still feel that this paper did not provide experimental evidence enough to prove field cancerization. I feel expression "find evidence for field cancerisation" is too affirmative. it should be weakened as an expression like "find a possibility of field cancerisation" .

We detail, below, our point-by-point response to any issues raised by the referees. We have also included the referees' comments in blue.

Reviewer #1:

This revised manuscript explores genomic and transcriptional changes in regions of apparently normal tissue that are distal and proximal to tumour lesions in 8 breast cancer patients. The revised manuscript is clearly written and much improved. Specifically, the authors have made a compelling case for field cancerisation as the mechanism drives the association between gene expression and proximity to the tumour (as opposed to paracrine signaling). They show that the proximity association module genes were more frequently mutated in cancers than those in modules not associated with proximity to the cancer lesion in several large data sets. They also provide functional evidence suggesting how some of the deregulated genes can contribute to carcinogenesis. The manuscript and the associated data will be of high interest to the breast cancer research community.

We thank Reviewer #1 for their Comments. No changes are requested.

Reviewer #2:

In the response to reviewer 2 the authors have addressed and satisfactorily answered questions and comments. The manuscript appears much improved and a bit more cautious in the conclusions of field cancerization. I think the manuscript deserves to be published and will represent a very valuable data set to continue to investigate.

We thank Reviewer #2 for their Comments. No changes are requested.

Reviewer #3:

Although the manuscript has been improved to some degree, I still feel that this paper did not provide experimental evidence enough to prove field cancerization. I feel expression "find evidence for field cancerisation" is too affirmative. it should be weakened as an expression like "find a possibility of field cancerisation" .

We thank Reviewer #3 for their Comments. In response, on line 60, we have replaced "find evidence for field cancerisation" with "find a possibility of field cancerisation". On lines 86 and 404, we have removed the word "strong".